

# Assessment and prediction of dust emissions, deposition and radiation forcing in Central Asia

Ying gan[123],Zhe Zhang[1234*], Wen Chu[5],Jianli Ding[6],Yuxin Ren[123]

1College of Geography and Remote Sensing Sciences, Xinjiang University, Urumqi, 830046, China

2 Xinjiang Key Laboratory of Oasis Ecology, Xinjiang University, Urumqi, 830046, China

3 Key Laboratory of Smart City and Environment Modelling of Higher Education Institute, Xinjiang University, Urumqi, 830046, China

4MNR Technology Innovation Center for Central Asia Geo-Information Exploitation and Utilization, Urumqi, 830046, China

5College of Marine Technology, Faculty of Information Science and Engineering;Ocean University of China;Qingdao;266100

6Xinjiang Institute of Technology, Aksu, China;

*    *Correspondence to: zhangzhe_0110@yeah.net;Tel:86+18799181249*

**Abstract.**    Dust aerosols significantly influence climate by modulating radiative balance and cloud processes. This study integrates MERRA-2 reanalysis data and the CMIP6 multi-model ensemble to assess the spatiotemporal evolution of dust emissions, deposition, and associated radiative effects in Central Asia from 1980 to 2100. Four SSP scenarios project that dust emissions in Central Asia exhibit a high-emission, high-deposition pattern with primary sources exceeding 15 $\mu g \cdot m^{-2} \cdot s^{-1}$. The deposition area significantly exceeds the source area (maximum >8 $\mu g \cdot m^{-2} \cdot s^{-1}$).Cross-scenario analysis demonstrates that dust emissions are highly sensitive to climate policy, with end-of-century emissions in the SSP5-8.5 high-emission scenario increasing by 94.9% relative to the baseline period. In contrast, emissions under the SSP1-2.6 low-carbon pathway vary by only 4.5%. Simulations using the SBDART model show that aerosol direct radiative forcing (ADRF) from dust in Central Asia under clear-sky conditions exhibits a vertical gradient, with cooling at the top of the atmosphere (TOA) and heating near the surface, yielding a net negative forcing at the TOA, with a minimum of <−10 W/m² near the Caspian Sea. Peak positive forcing within the atmosphere, observed in spring, reaches 10.0 W/m². Increased dust emissions reduce shortwave radiation at the surface by up to −20 W/m². Ground-based observations indicate seasonal variations in the dust-induced heating rate, with peak near-surface heating in spring at Kashgar (93.0 W/m²) and a maximum heating rate of 2.6 K/day. In contrast, the near-surface heating rate at Issyk-Kul Lake in autumn (0.34 K/day) is approximately four times higher than in spring (0.08 K/day).



**Graphical Abstract**

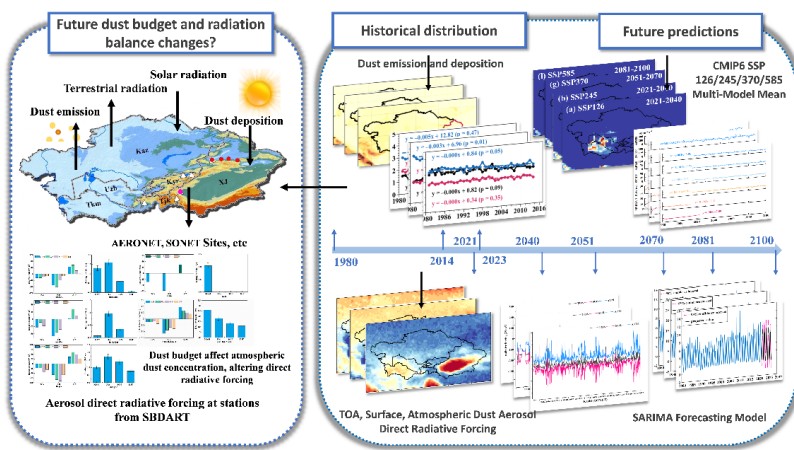

**Keywords:** Dust cycle;CMIP6 multi-model ensemble(MME); Direct radiation forcing of dust;
SBDART model
**1.introduction**
Dust aerosols are a significant component of the mass load of tropospheric aerosols, accounting for up
to approximately 50%, and have a profound impact on the functioning of the Earth system (Mahowald
et al. 2010, Ramanathan et al. 2001). Their trans-circulation process (lithosphere-atmosphere-cryosphere)
and interaction with the climate system have become cutting-edge research areas in Earth system science.
The release, transport, and deposition of dust aerosols not only involve multiple geospheres but also have
a significant impact on weather and climate, air quality, and even human health after entering the
atmosphere (Tegen et al. 2004, Penner et al. 2006, Pozzer et al. 2012).
Global annual dust emissions are enormous, ranging from approximately 1000 to 2150 Tg, with 30% to
40% originating from arid regions of Asia (Tanaka and Chiba 2006). Dust is transported across continents
by the westerly wind circulation, which significantly impacts the atmospheric radiation balance in East
Asia, North America, and even the Arctic region (Wallace and Hobbs 2006). Although studies have
confirmed that dust regulates the land-atmosphere energy budget through direct radiative forcing
(including scattering and absorption of short-wave and long-wave radiation) and indirect effects (such as
changing the efficiency of precipitation as cloud condensation nuclei), significant uncertainty remains
regarding the vertical distribution of dust, the amplification mechanism of anthropogenic emissions, and



its regional climate feedback (IPCC AR6).
Due to the challenges associated with dust observation, our understanding of the behavior of dust
throughout its life cycle remains insufficient, hindering a complete understanding and accurate modeling
of its complex mechanism of action (Kok et al. 2023). Numerous studies have used a variety of methods,
including in situ observations, satellite remote sensing, and model simulations, to thoroughly examine
the spatiotemporal changes, optical properties, and radiative forcing of dust aerosols(Wang et al. 2018,
Song et al. 2021, Chen, Zhao and Fan 2022). For example, global dust are primarily confined to the "dust
belt," with approximately one-third originating from the Asian region(Kok et al. 2023). Dai et al. utilized
a variety of remote sensing and ground-based data to study the sources, microphysical characteristics,
and optical properties (Dai et al. 2022, Salvador et al. 2022). Zhao et al. investigated the simulation of
global and regional dust by 16 CMIP6 models in the Atmospheric Model Intercomparison Project (AMIP)
experiment and compared the results with observational and reanalysis data (Zhao et al. 2023, Liu et al.

62    2024).

Mode simulation provides information on the temporal and spatial changes of dust aerosols worldwide
and helps predict future trends (Li et al. 2021). The results of climate models such as CMIP5 and CMIP6
have enabled us to understand the main characteristics of dust aerosols better. These models have
increasing resolutions and increasingly complex physical processes and parameterizations,
demonstrating their ability to simulate dust events and processes on meso- to global scales(Zhao, Ryder
and Wilcox 2022). In particular, the CMIP6 experiment has provided a valuable opportunity to
understand the impact of dust emissions on climate and the role of dust in the latest generation of climate
models (Braconnot et al. 2021, Zhao, Wilcox and Ryder 2024).
The flux of dust emissions(Shen et al. 2016). However, most current research focuses on the
spatiotemporal distribution and transport processes of dust(Li et al. 2022, Tao et al. 2022). A systematic
understanding of key aspects of the local dust life cycle in this region remains lacking, including the
long-term evolution of the dust emission-deposition budget, the high dependence of the vertical dust
profile on direct radiative forcing, and the modal differences in the climatic feedback of dust under
different carbon emission scenarios. These deficiencies in understanding seriously limit the reliability of
climate models in Central Asia, with the uncertainty in radiation-forcing estimates primarily stemming
from insufficient ground verification due to a lack of sites (Brown et al. 2021, Wu and Boor 2021).



To overcome the aforementioned bottleneck, this study establishes a multi-source data fusion framework
that seamlessly integrates the following observational and modeling resources: (1) MERRA-2 reanalysis
data and the CMIP6 multi-model ensemble are employed to construct high-resolution data on the dust
budget in Central Asia through dynamic downscaling and forecast the evolutionary trend of the climatic
effect of dust under different scenarios; (2) The SBDART radiative transfer model is integrated with the
valuable measured data from the SONET Asian Dust Observation Network and the Jinghe CE318
ground-based remote sensing station to quantify the long-term trend of the shortwave radiation forcing
(ADRF) of dust under clear-sky conditions; (3) The SARIMA statistical model is used to predict the
short-term evolutionary trend of the radiative effect of dust. This method is the first to achieve a closed-
loop analysis of the entire "emission-deposition-radiation" chain of dust in the source region using
multiple datasets, providing key constraints for elucidating the physical mechanisms of the dust life cycle.
The structure of this paper is as follows. Section 2 presents the data sources, the downscaling method for
the CMIP6 dust budget, and the calculation method for clear-sky aerosol radiative forcing. Section 3
examines the detailed characteristics of the dust budget, projections of future changes, and the radiative
forcing of dust aerosols. Finally, the main conclusions and a discussion are presented in Section 4.
**2. Data and Methods**
**2.1 Data sources**
The study area is situated between 35°-57°N and 48°-96°E, encompassing the five Central Asian
countries (Kazakhstan, Uzbekistan, Tajikistan, Turkmenistan, and Kyrgyzstan) and the Xinjiang region
of China (comprising both its northern and southern parts). This region is positioned in the hinterland of
the Eurasian continent and is characterized by a temperate continental climate with extreme aridity. The
region features a highly heterogeneous surface, with the Taklamakan Desert and the surrounding Gobi
(comprising over 40% of the study area) interspersed with mountain ranges, such as the Tianshan and
Pamir, forming a unique landform (Shen et al. 2016, Hetzel et al. 2002). As the world's second-largest
source of dust, strong thermal and dynamic coupling drives intense dust activities (Zhang et al. 2020),
with emission hotspots concentrated in the Tarim Basin, the dried bed of the Aral Sea, and the Kazakh
steppe belt. This study focuses on the regional dust budget and radiative effects, utilizing MERRA-2



reanalysis data, the CMIP6 multimodel ensemble, AERONET, SONET, and hand-held photometer data.

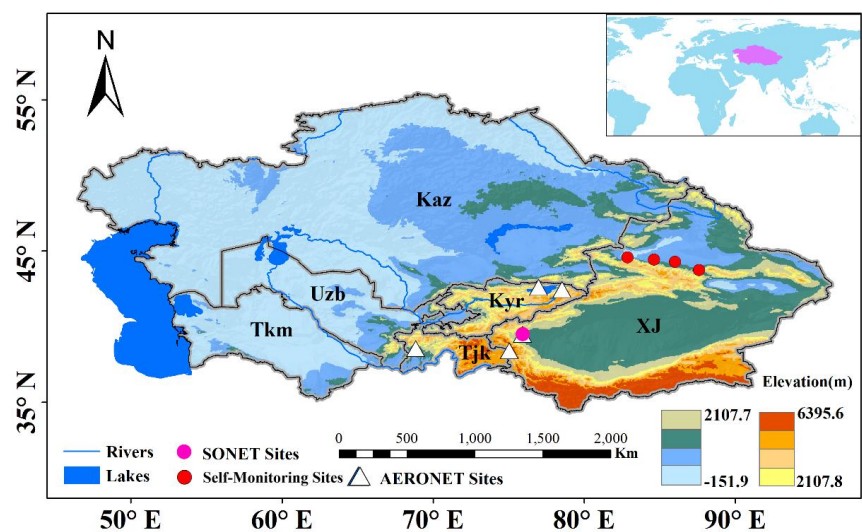

**Figure 1  Location of the Study area.**
**2.1.1 Data from ground-based heliometers**
AERONET (AErosol RObotic NETwork) employs a CE-318 solar photometer to measure aerosol optical
depth (AOD) across 8 bands in the range of 340–1640 nm and to derive microphysical parameters,
including single scattering albedo (SSA), refractive index (m), and particle size spectrum (Holben et al.
1998, Holben et al. 2001). The Level 2 data exhibit an uncertainty of less than 5%. As an internationally
recognized standard for ground-based aerosol observations, its long-term stability and algorithmic
consistency offer a reliable input for radiative forcing calculations (García et al. 2012).
The Chinese Academy of Sciences-led SONET (Sun-sky radiometer Observation NETwork) employs
the CE318-DP instrument to provide information on the chemical composition and vertical profile of
aerosols while adhering to the stringent quality control procedures of AERONET. The establishment of
SONET sites has effectively addressed the gaps in AERONET's spatial coverage in this source region
(Li et al. 2018). Cross-validation demonstrated that the correlation coefficient between SONET and
AERONET AOD was 0.98 (RMSE < 0.02), confirming a seamless connection between the two data sets
(She et al. 2024).
To supplement the insufficient temporal and spatial coverage of fixed stations, this study employs CE-



318 and Microtops II handheld photometer to obtain transient AOD observations in the 550–870 nm
band (accuracy ±0.01) for verifying the local applicability of satellite inversion products. By integrating
the aforementioned multi-scale observational data, this paper employs AERONET and SONET Level 2
data to provide vertical profiles of the optical-physical properties of aerosols, calculate the direct radiative
forcing of aerosols, and validate the satellite data on AOD and radiation flux in Central Asia
(Supplementary Figure 1).
**2.1.2 MERRA-2 reanalysis data**
The MERRA-2 reanalysis data used in this study was developed by the NASA Goddard Space Flight
Center. Its core is based on the GEOS-5 atmospheric circulation model and the ADAS-5.12.4
assimilation system. A global multi-element dataset with 72 vertical layers (surface to 80 km) and a
horizontal resolution of 0.625°×0.5° has been constructed from 1980 to the present by fusing satellite
remote sensing (MODIS/AVHRR aerosol optical thickness), ground-based observations (soundings,
aircraft observations), and the GOCART aerosol chemical transport model output (Gelaro et al. 2017).
In addition to covering variables related to cloud, radiation, and hydrological cycles, the coupled
GOCART model distinguishes the interaction mechanisms of five types of aerosols in this dataset: dust
(DU), sea salt (SS), sulfate ($SO_4$), black carbon (BC), and organic carbon (OC). For the first time, the
entire life cycle of dust aerosols has been analyzed, providing key parameters such as monthly average
dust emission flux, dry/wet deposition rate, particle size-classified loads, and single scattering albedo at
483.5 nm, ensuring physical consistency for quantifying the radiative forcing of dust (Buchard et al.
2017). Based on the advantages of this data, this study extracts radiation flux and dust cycle parameters
under clear sky conditions in Central Asia and systematically constructs a collaborative analysis
framework for dust emissions, deposition, and radiative forcing.
**2.1.3 CMIP6 model simulations**
The Sixth Coupled Model Intercomparison Project (CMIP6) integrates 112 climate models from 33
institutions worldwide, with its multi-scenario simulation significantly surpassing previous studies in
both breadth and depth. To analyze the interdecadal variations in dust emissions and wet and dry
deposition in Central Asia, 10 models were selected from CMIP6 based on the principle of data





completeness (Eyring et al. 2016). The selection criteria include key variables of the dust cycle: monthly
mean dust emission fields and dry/wet deposition fluxes from 1980 to 2014 for the historical period, and
from 2015 to 2100 for four shared socioeconomic pathways (SSP1-2.6, SSP2-4.5, SSP3-7.0, and SSP5-

154 8.5).

To ensure spatial consistency in the comparison of multi-source data, all model output data were
subjected to statistical downscaling and aligned with MERRA-2 reanalysis data (spatial resolution
0.625°×0.5°). This multi-model ensemble effectively characterizes the uncertainty in climate responses
while controlling computational costs, providing reliable data support for analyzing the long-term
evolution of the dust cycle in the arid region of Central Asia.
**Table.1 Overview of the models and simulations used in this study.**

| Model | Nation | Resolution | Hist | SSP126 | SSP245 | SSP370 | SSP585 | Dust emission scheme | Model references |
|---|---|---|---|---|---|---|---|---|---|
| CESM2-WACCM | USA | 1.25°×0.94° | 3 | 1 | 5 | 3 | 5 | Zender et al. (2003) | Danabasoglu et al. (2020) |
| CESM2 | USA | 1.25°×0.95° | 11 | 3 | 3 | 3 | 3 | Zender et al. (2003) | Wu et al. (2016) |
| CNRM-ESM2-1 | France | 1.25°×0.96° | 3 | 5 | 10 | 5 | 5 | Marticorena et al. (1997) | Séférian et al. (2019) |
| GFDL-ESM4 | USA | 1.25°×0.97° | 1 | 1 | 1 | 1 | 1 | Evans et al. (2016) | Dunne et al. (2020) |
| GISS-E2-1-G | USA | 1.25°×0.98° | 19 | 10 | 25 | 17 | 10 | Ginoux et al. (2004) | Bauer et al. (2020) |
| GISS-E2-1-H | USA | 1.25°×0.99° | 10 | 5 | 5 | 1 | 5 | Bauer and Koch.(2005) | Kelley et al. (2020) |
| GISS-E2-2-G | USA | 1.25°×1.00° | 5 | 5 | 5 | 5 | 5 | Cakmur et al. (2006) | Rind et al. (2020) |
| MRI-ESM2-0 | Japan | 1.25°×1.01° | 12 | 5 | 10 | 5 | 6 | Tanaka and Chiba.(2005) | Yukimoto et al. (2019) |
| HadGEM3-GC31-LL | UK | 1.875°×1.25° | 5 | 3 | 4 | 2 | 3 | Marticorena. (1995) | Williams et al. (2020) |
| UKESM1-0-LL | UK | 1.25°×0.103° | 3 | 5 | 5 | 3 | 4 | Marticorena. (1995) | Senior et al. (2020) |



**2.2 Methodology**
**2.2.1 Delta statistical downscaling**
Due to the limited original spatial resolution of the CMIP6 model (with a typical horizontal grid of
approximately $1.25° \times 1°$), direct application to the analysis of regional-scale dust cycles may introduce
systematic biases. Therefore, the Delta Change Factor method is employed in this study for statistical
downscaling. The core of this method is to separate the historical bias of the climate model from the
future change signal, followed by the reconstruction of high-resolution climate elements (Maraun et al.
2010, Gutmann et al. 2014).
First, the baseline period deviation is calculated, and the monthly mean dust emission $P_{m,his}$ of the
historical simulation data (1980-2014) of each CMIP6 model is extracted. The grid is then matched with
the MERRA-2 reanalysis observations $P_{obs}$ for the same period to calculate the model's systematic
deviation ratio:
$B_m = \frac{P_{m,his}}{\overline{P_{obs}}}$     (1)
where $\overline{P_{obs}}$ is the monthly average of the observation period, and $Bm$ represents the spatial deviation of
model m in the reference period.
Second, the relative change factor for future scenarios is extracted, and the ratio of dust emissions for
each model during the future scenario period (2015-2100) relative to its own historical simulation is
calculated.
$R_{m,fut} = \frac{P_{m,fut}}{P_{m,his}}$     (2)
Among them, $P_{m,fut}$ is the average monthly emission of mode m in the future, and $P_{m,his}$ is the average
monthly emission of mode m over multiple years in the historical period.
This method decouples the historical deviation from the climate change signal, retaining the physical
response characteristics of CMIP6 to future climate forcing while improving the simulation accuracy at
the regional scale through the use of high-resolution observational data. Compared with dynamic
downscaling, it significantly reduces computational costs and is well-suited for multi-model uncertainty
quantification research.



**2.2.2 SBDART Radiative Transfer Model Calculation of Direct Radiative Forcing of Aerosols**
The SBDART radiation transfer model (Ricchiazzi et al. 1998) was employed in this study to
quantitatively assess the direct radiative effect of aerosols. SBDART solves the atmospheric radiation
transfer equation using the four-stream approximation method. Its core architecture comprises three
modules: First, the discrete ordinates radiation transfer (DISORT) module calculates the radiative fluxes
of the 45-layer atmosphere (with a vertical resolution of 0.3 km); second, the spectral parameterization
module integrates the LOWTRAN-7 atmospheric absorption spectrum and Mie scattering theory to cover
the shortwave band from 0.25 to 4.0 μm; and finally, the surface-atmosphere coupling module analyzes
the radiative interaction between surface albedo and atmospheric constituents such as water vapor and
ozone.
Our research is based on a comprehensive data collection, with key input parameters including the optical
properties (e.g., optical depth τ, single scattering albedo SSA, asymmetry factor ASY), and the vertical
profiles of aerosols. These are obtained from the solar photometer observation network at the Central
Asia site, which offers a significant advantage in temporal and spatial resolution over satellite retrieval
products (Dubovik and King 2000). To quantify the radiative forcing of dust aerosols, all simulations
were performed under clear sky conditions, and the solar zenith angle was constrained according to the
seasonal average value of the study area to ensure the comparability of regional radiative effects
(Halthore et al. 2005). The aerosol direct radiative forcing (ADRF) is calculated using the standard
scientific approach to determine the difference in net radiative flux with and without aerosols under
cloud-free conditions. Specifically, the ADRF at a given altitude z, at the top of the atmosphere (TOA),
at the surface (SFC), and in the atmosphere (ATM), can be defined as follows:
$NF_z = F_{z,down} - F_{z,up}$ (3)
$ADRF_z = NF_z^{aer} - NF_z^{noaer}$ (4)
$ADRF_{TOA} = NF_{TOA}^{aer} - NF_{TOA}^{noaer}$ (5)
$ADRF_{SFC} = NF_{SFC}^{aer} - NF_{SFC}^{noaer}$ (6)
$ADRF_{ATM} = ADRF_{TOA} - ADRF_{SFC}$ (7)
$ADRF_{dust} = ADRF \times \left(\frac{DAOD}{AOD}\right)$ (8)
Among them, $F_{z,down}$ and $F_{z,up}$ are the downward and upward radiative fluxes, $NF_z^{aer}$ and $NF_z^{noaer}$ are
the net radiative fluxes with and without aerosols, and ADRF is the aerosol direct radiative forcing.



### 2.2.3 SARIMA prediction model

Given the non-stationarity and interannual cycle characteristics of the radiation forcing time series of Central Asian dust, this study employs the seasonal autoregressive integrated moving average model (SARIMA) for modeling and analysis. First, the augmented Dickey-Fuller test (ADF, p < 0.05) was used to identify the non-stationarity of the series. A compound differencing strategy (first-order conventional difference d=1, first-order seasonal difference D=1, period s=12) was employed to eliminate the trend and interannual fluctuations, resulting in a stationary residual series (KPSS test p > 0.1).

The non-seasonal order (p=2, q=1) was determined based on the autocorrelation function (ACF) and partial autocorrelation function (PACF), while the seasonal order (P=1, Q=1) was optimized using grid search, resulting in the final SARIMA(2,1,1)(1,1,1)$_{12}$ model (AIC=112.3, BIC=125.7). Model validation demonstrated that the goodness of fit was R²=0.87 for annual cycle dynamics, and the prediction error for extreme event peaks was less than 15%, confirming its effectiveness in the analysis of non-stationary sequences (Sirisha, Belavagi and Attigeri 2022).

### 3. Results and analysis

### 3.1 Spatial pattern and multimode prediction of dust emissions in Central Asia

Figure 2 compares MERRA-2 observations with CMIP6 multi-model ensemble (MME) dust emissions from 1980 to 2014. The historical spatial distribution of the ten models is shown in Supplementary Figure 2. The reanalysis data agrees with the MME simulations, with the Taylor skill score (SS) close to 1. Dust emissions in the study area exhibit significant temporal and spatial variation. In terms of spatial distribution (Figure 2a), both datasets consistently identify the three primary core emission sources in the Tarim Basin, the dried-up Aral Sea area, and the Gobi Desert, with maximum emission fluxes exceeding 15 µg·m⁻²·s⁻¹. Regarding the trend of dust loads (Figure 2b), dust emissions in the Aral Sea region have increased significantly (>0.5) over the past 34 years, while those in the Tarim Basin have slightly decreased (≈-0.3).

The Aral Sea region has experienced a 68% reduction in lake area since 1960, resulting in 54,000 km² of exposed lakebed (Wang et al. 2020). Under arid climatic conditions with an annual average precipitation of less than 100 mm and potential evaporation of more than 2000 mm, the dust emission flux has increased significantly at a rate of approximately 0.5 µg·m⁻²·s⁻¹·yr⁻¹ over 34 years. In contrast, the Tarim



Basin has benefited from ecological restoration projects and increased precipitation during the growing
season (Fu et al. 2021), leading to a decrease in emission flux at a rate of ≈-0.3 μg·m⁻²·s⁻¹·yr⁻¹. Time
series analysis (Figure 2c) shows that overall dust emissions fluctuate gently without significant annual
changes. Dust emissions in the southern Tarim Basin of Xinjiang increase and decrease annually,
consistent with the spatial distribution map of the trend. Dust emissions in northern Xinjiang are similar
to those in Central Asia, with northern Xinjiang slightly higher than other regions in Central Asia. This
may be attributed to local differences in surface roughness and land use, reflecting regional disparities in
emission characteristics.

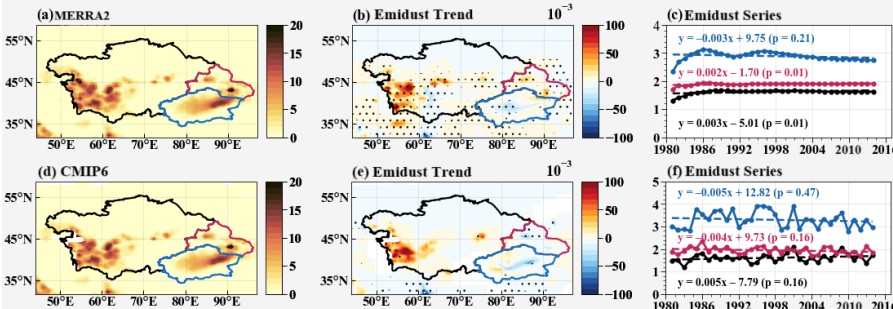


**Figure. 2  Spatial distribution, linear trend, and time series of dust emissions from MERRA-2 and CMIP6**
**MME in Central Asia from 1980 to 2014. Red highlights the northern part of Xinjiang, blue indicates the**
**southern part of Xinjiang, and black indicates the five Central Asian countries. The black dots in (b) and (e)**
**indicate the 90% confidence level regions.**

Figure 3 illustrates the relative changes in Central Asian dust emissions in the near term (2021–2040),
medium-term (2051–2070), and long-term (2081–2100), compared to the reference period (1980–2014),
with significant temporal and spatial differences observed. In all scenarios, areas with high dust values
(>50 μg·m⁻²·s⁻¹) are consistently distributed across the Aral Sea hinterland, Turkmenistan, and the
eastern edge of the Tarim Basin. Dust emission intensity is positively correlated with the radiative forcing
scenario, with intensity continuing to increase over time within a given scenario (long term > medium
term > short term). Specifically, short-term emissions in the Aral Sea region range from 17.8 (SSP370)
to 26.0 (SSP245) μg·m⁻²·s⁻¹, with relatively minor differences. However, in the long term, under the
high-radiation scenario (SSP585), emissions surge to 387.1 μg·m⁻²·s⁻¹, representing a 94.9% increase
compared to the reference period. This sharp increase is directly attributed to the exposure of saline
sediments on the lake bed, soil loosening due to rising surface temperatures, and increased wind erosion
(Lioubimtseva and Cole 2006). In contrast, the Tarim Basin experienced a long-term decrease in



emissions ranging from 18.7% (SSP245) to 29.3% (SSP370) due to ecological restoration (with a 10-
year NDVI increase of 0.12) and an increase in growing season precipitation (Xu et al. 2019). The
SSP585 scenario shows a decrease from 27.2 (short-term) to 20.1 $\mu g \cdot m^{-2} \cdot s^{-1}$, reflecting a 26.1%
reduction. Regional comparisons reveal significant differences in the sensitivity of climate responses.
Emissions from the Aral Sea increase exponentially with the intensity of radiative forcing ($R^2 = 0.93$).
Meanwhile, the southern Xinjiang region shows a gradual decreasing trend, confirming the potential of
human intervention in regulating the dust process.

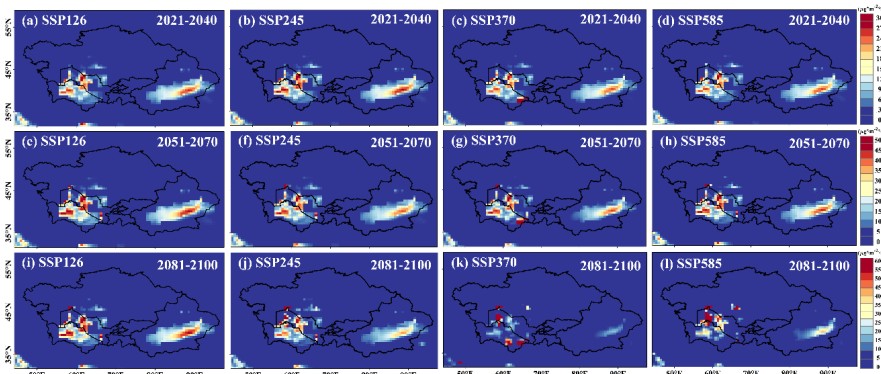


**Figure.3  Future changes in dust emissions over time. Spatial variability of dust emissions in Central Asia**
**under the four SSP scenarios of CMIP 6 MME for (a-d) the near term (2021-2040), (e-h) the medium term**
**(2051 - 2070), and (i-l) the long term (2081 - 2100) relative to the historical period (2000-2014).**
**3.2 Spatial pattern and multi-model prediction of dust deposition in Central Asia**
Dust emissions and deposition together constitute the complete dust budget, with deposition serving as
the final outcome of emissions. Dust particles entering the atmosphere are redistributed at the surface-
atmosphere interface through gravity-dominated dry deposition and precipitation-driven wet deposition
(see Supplementary Figures 3-4 for the historical distribution of dry and wet deposition across 10 models)
(Marticorena and Bergametti 1995, Shao et al. 2011). Figure 4 shows that the multimodel ensemble
(MME) and the observed data are in good agreement in simulating total dust deposition in Central Asia,
though the deposition trend observed by MERRA-2 is significantly stronger than that of the model
ensemble. In terms of spatial distribution, the high-value sedimentation areas (>5 $\mu g \cdot m^{-2} \cdot s^{-1}$) heavily
overlap with emission hotspots, concentrated in the western part of Central Asia and the Tarim Basin in
southern Xinjiang, confirming the local coupling mechanism of dust "generation-deposition." The trend
analysis (Figure 4b) shows that the eastern edge of the Aral Sea and the East Caspian Sea exhibit the



most substantial positive trend ($\Delta S$ = +0.15 μg·m⁻²·s⁻¹). Meanwhile, southern Xinjiang is characterized
by a negative trend ($\Delta S$ = -0.10 μg·m⁻²·s⁻¹). Time series analysis (Figure 4c) indicates that the observed
data for the period 1980-2014 exhibit a slowly increasing trend in sedimentation flux in Central Asia,
with an annual rate of change of 0.002. In contrast, the MME simulation for the Xinjiang region shows
a slight decrease of -0.003 yr⁻¹. This discrepancy between observations and simulations may arise from
the model's uncertainty regarding the boundary layer's dynamic processes and the parameterization of
precipitation microphysics in Central Asia's arid region. Specifically, the quantification of wet deposition
efficiency for dust requires further improvement.

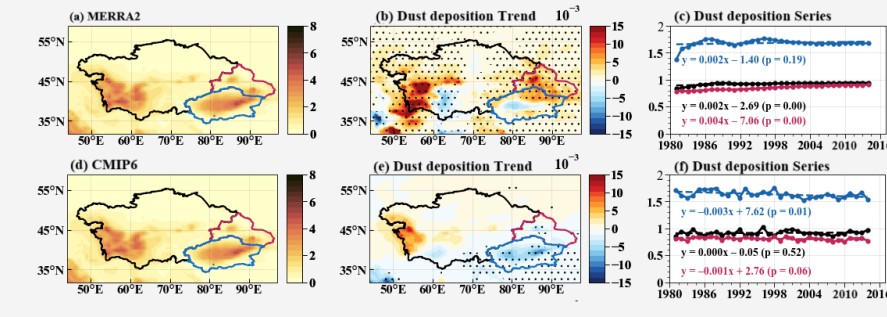

**Figure. 4  Spatial distribution, linear trend, and time series of dust deposition (dry deposition + wet**
**deposition) for MERRA-2 and CMIP6 MME in Central Asia from 1980 to 2014. Red highlights the northern**
**part of Xinjiang, blue indicates the southern part of Xinjiang, and black indicates the five Central Asian**
**countries. The black dots in (b) and (e) indicate the regions with a 90% confidence level.**
Figure 5 illustrates the relative changes in dust deposition projections over time for the four scenarios
(see Supplementary Figures 5-6 for future changes in dry and wet deposition). In contrast to the
distribution of the dust emission source area, the influence of deposition extends to the surrounding
regions, primarily covering the southwestern part of Central Asia, the southeastern edge of the Tarim
Basin, and the Junggar Basin, with the maximum deposition flux exceeding 8 μg·m⁻²·s⁻¹, presenting a
spatial pattern of "deposition domain > emission source." In terms of temporal evolution, the mean values
ranged from 9.3 (SSP585) to 10.4 (SSP245) μg·m⁻²·s⁻¹ in the near term (2021-2040) and from 9.6
(SSP370) to 10.0 (SSP126) μg·m⁻²·s⁻¹ in the long term (2081-2100), with an overall variation of less
than 12%. This phenomenon may stem from the compensatory effect of wet and dry deposition processes,
with dry deposition fluxes decreasing by approximately 0.2% per year under medium- to high-level
radiative forcing in the southern Xinjiang region due to precipitation patterns. In contrast, dry deposition
increases in western Central Asia due to the enhancement of near-surface winds resulting from reduced





surface roughness. However, the spatial and temporal stability of the wet deposition process mitigates
the overall deposition fluctuations.

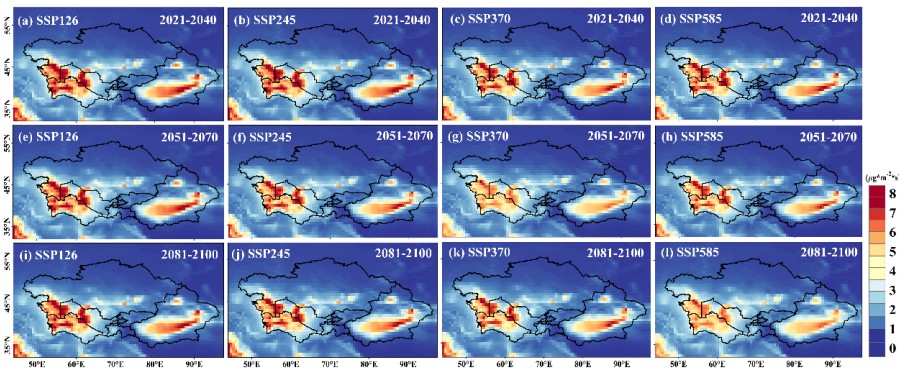

**Figure. 5  Future changes in dust deposition over different periods. Spatial variability of dust deposition in**
**Central Asia under the four SSP scenarios of CMIP 6 MME for (a-d) the near term (2021-2040), (e-h) the**
**mid-term (2051 - 2070) and (i-l) the long term (2081 - 2100) relative to the historical period (2000-2014).**
In order to more accurately assess the trend simulation performance of the dust cycle, we constructed a
time series of dust emission and wet and dry deposition between 1980 and 2100 based on the MERRA-
2 and CMIP6 multimodel ensembles (MME) (see Fig. 6). Overall, the simulation results show that dust
emissions in Xinjiang remain relatively stable over the next 120 years. In contrast, in the five Central
Asian countries, especially under scenarios with higher radiative forcing (e.g., SSP370, SSP585), dust
emissions increase significantly between 2081 and 2100, accompanied by a fluctuating gradual increase.
In contrast, dust deposition (both dry and wet deposition) shows a smoother trend with less volatility.
In the specific analyses, dust emissions from MERRA-2 exhibited a smooth trend, with an average of 30
$\mu g \cdot m^{-2} \cdot s^{-1}$ in the Tarim Basin and 15 $\mu g \cdot m^{-2} \cdot s^{-1}$ in other regions. In contrast, simulated data from the
multi-model ensemble exhibited slight fluctuations, with peaks exceeding 45 $\mu g \cdot m^{-2} \cdot s^{-1}$ at certain times.
Some deviation is observed in the temporal variability between the two. The volatility of dry deposition
of dust is relatively small, with a slope of less than 0.1, indicating that the dry deposition process is
smooth. Additionally, both dry and wet deposition do not exhibit significant volatility in the long-term
trend. Wet deposition exhibited slight deviation in northern Xinjiang and remained relatively smooth in
other regions. The average wet deposition flux was approximately 1.5 $\mu g \cdot m^{-2} \cdot s^{-1}$, with an overall slope
of less than 0.2, indicating relatively small variation. Notably, the wet deposition data from MERRA-2
exhibited a significant increase in the northern border region around 2000. This change may be related
to the assimilation of MODIS satellite and other observations by MERRA-2. Therefore, MERRA-2 data





from 2000 to 2014 were selected to calibrate the model and ensure the accuracy of the simulation results.
In summary, although future dust emissions vary significantly under different climate scenarios, the
overall dust deposition process remains relatively stable. The simulation results from MERRA-2 and the
multi-model ensemble exhibit spatial and temporal differences in different regions.

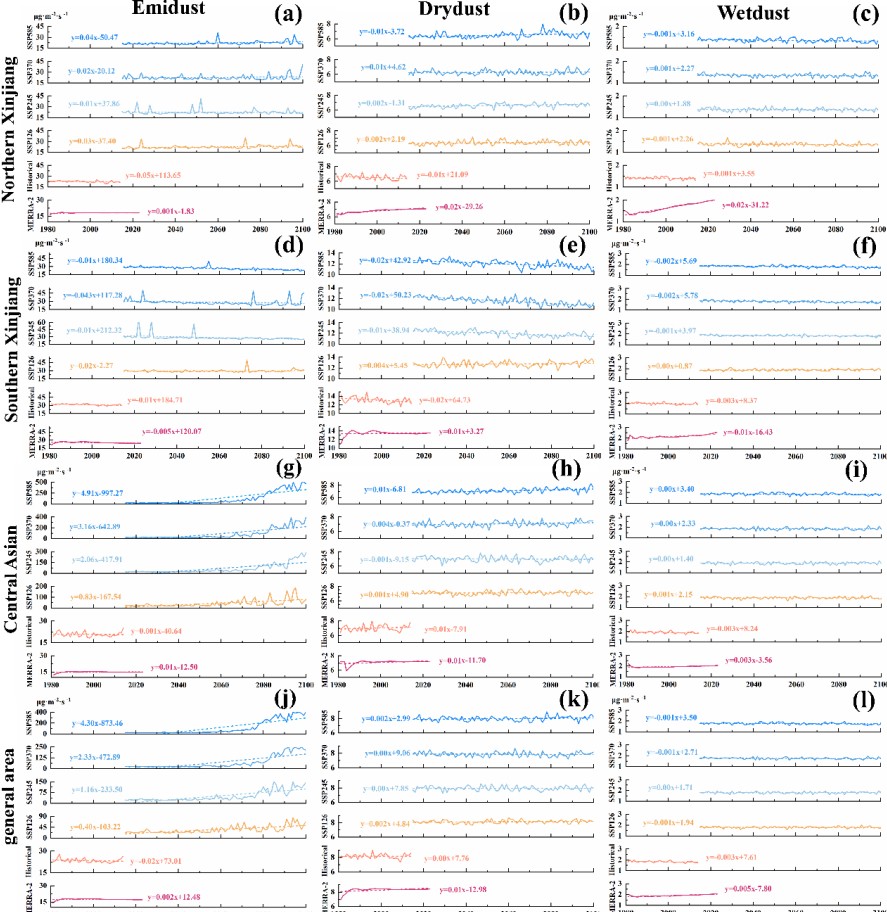


**Figure. 6  Time evolution of dust receipts and payments. Dust emissions, and wet and dry deposition (µg-m-**
**²-s-¹) in the northern (a-c), southern (d-f) Central Asia (g-i), and the whole region (j-l). From CMIP 6 MME**
**1980-2100, MERRA-2 results 1980-2023.**
**3.3 Analysis of changes in direct radiative forcing of dust aerosols**
**3.3.1 Monthly average changes in direct radiative forcing by dust aerosols**
Based on the above quantitative characterization of dust emission sources and deposition processes,
further investigation is required to elucidate the perturbation mechanism of dust aerosols on the energy



balance of the surface-atmosphere system. This study quantifies the radiative balance impacts of Central
Asian dust aerosols at different spatial and temporal scales through the short-wave direct radiative forcing
(ADRF) using MERRA-2 observations of dust aerosols under clear-sky conditions from 1980 to 2023.
As shown in Fig. 7a-d, the top-of-atmosphere (TOA) radiative forcing exhibits considerable spatial
heterogeneity. The overall negative forcing is characterized by the lowest value (<-10 W/m²) in the
Caspian Sea region, followed by the Tarim Basin and the Aral Sea region (<-8 W/m²), confirming that
dust aerosols exert a significant cooling effect by enhancing shortwave reflection. The seasonal analysis
showed that the negative TOA forcing intensity followed a decreasing pattern: spring (-3.32 W/m²) >
summer (-3.21 W/m²) > fall (-3.07 W/m²) > winter (-1.94 W/m²), which was closely associated with the
seasonal characteristics of dust activity.
The spatial pattern of surface (SFC) radiative forcing (Fig. 7e-h) shows a more pronounced negative
distribution, with two cooling centers forming in the Tarim Basin and southwestern Central Asia, peaking
at shortwave radiation losses of -20 W/m². This phenomenon results from the synergistic effect of
scattering and absorption within the atmosphere, a dual attenuation mechanism of incident solar radiation
by dust particles (Li et al., 2022a), which significantly diminishes the surface energy balance. Notably,
the atmospheric radiative forcing (ADRF) is consistent with TOA and SFC in spatial distribution.
However, its positive characteristics (10.02 W/m² in spring and 9.89 W/m² in summer) reveal the energy
redistribution role of dust aerosols, which attenuate surface radiation while trapping energy in the
atmospheric system through absorption processes.

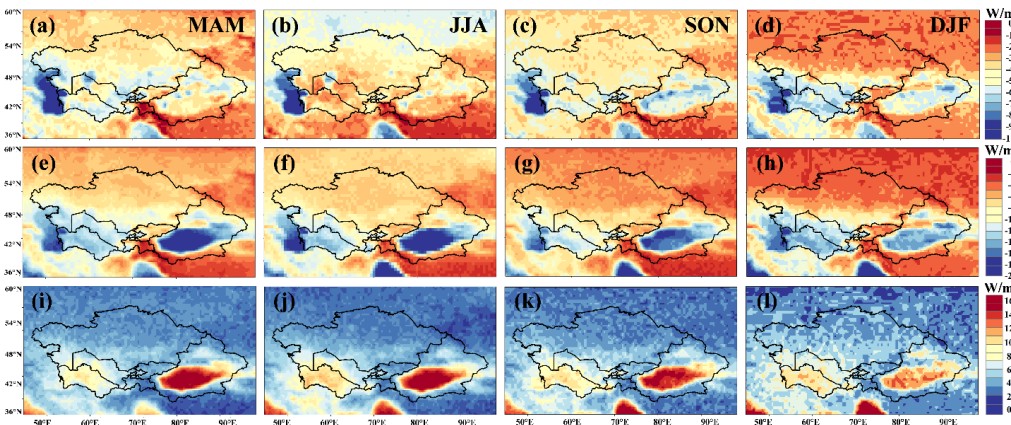

**Figure. 7 Seasonal spatial distribution of direct radiative forcing of dust aerosols in Central Asia, 1980-2023,**
**taking into account clear-sky shortwave aerosol direct radiative forcing at the top of the atmosphere (a-d), at**
**the surface (e-h) and in the atmosphere (i-l).**



**3.3.2 Refinement of aerosol direct radiative forcing in dusty weather**


Following an in-depth exploration of the spatial distribution characteristics of dust aerosol direct
radiative forcing in the atmosphere, as revealed by MERRA2 reanalysis data, this study was further
refined to investigate the temporal divergence characteristics of the direct radiative effect of dust aerosols
and its physical mechanisms at typical sites in Central Asia, in conjunction with the SBDART
atmospheric transport model (Fig. 8). The dynamic response of site-scale aerosol direct radiative forcing
(ADRF) to the atmospheric heating rate is quantitatively analyzed.
Observations indicate that seasonal changes characterize ADRF. The Dushanbe, Issyk-Kul, and Jinghe
sites exhibit peak radiative forcing in summer (56.72, 34.22, and 61.17 W/m²) and subsequently decrease
to the annual minimum in winter (approximately 2.33 W/m² in Dushanbe and 27.36 W/m² in Jinghe,
respectively). This pattern coincides with the period of frequent dust occurrence during summer in
western Central Asia, influenced by the westerly wind circulation(Li et al. 2022). Notably, the Kashgar
site exhibits a unique spring-dominant pattern (92.99 W/m²), which may be attributed to the synergistic
effect of surface exposure following spring snowmelt and vigorous Mongolian cyclone activity, along
with the unique sand initiation mechanism in the Tarim Basin.
Changes in the atmospheric heating rate maintain a significant positive correlation with ADRF,
confirming the central role of radiation absorption by dust aerosols. The peaks in heating rates at all
stations occur during the active dust period: Dushanbe (1.29 K/day in summer) and Jinghe (1.72 K/day
in summer) align with the westerly transport paths, while the anomalously high value in spring at Kashgar
(2.61 K/day) corresponds to the significant sand uplift event in the Taklamakan Desert. It is noteworthy
that the heating rate at Issyk-Kul in spring (0.08 K/day) is significantly lower than that in fall (0.34
K/day), which may be related to the site being shielded by mountainous terrain, limiting vertical transport
of dust in spring. This may also impact the accuracy of the results, considering the relative scarcity of
observational data at the Issyk-Kul site.





In this study, it was found that the spatial and temporal divergence of regional radiative effects is
primarily controlled by two major factors: (1) seasonal modulation of emission intensity in dust source
regions, such as the enhanced transport of dust from the westerly rapids to the Aral Sea basin in summer,
and (2) modulation of localized atmospheric boundary layer processes, typically manifested as the
difference in thermal response between a mountainous site (Issyk-Kul) and a basin site (Kashgar). These
findings provide essential observational constraints for improving the dust-radiation parameterization
scheme in regional climate models.

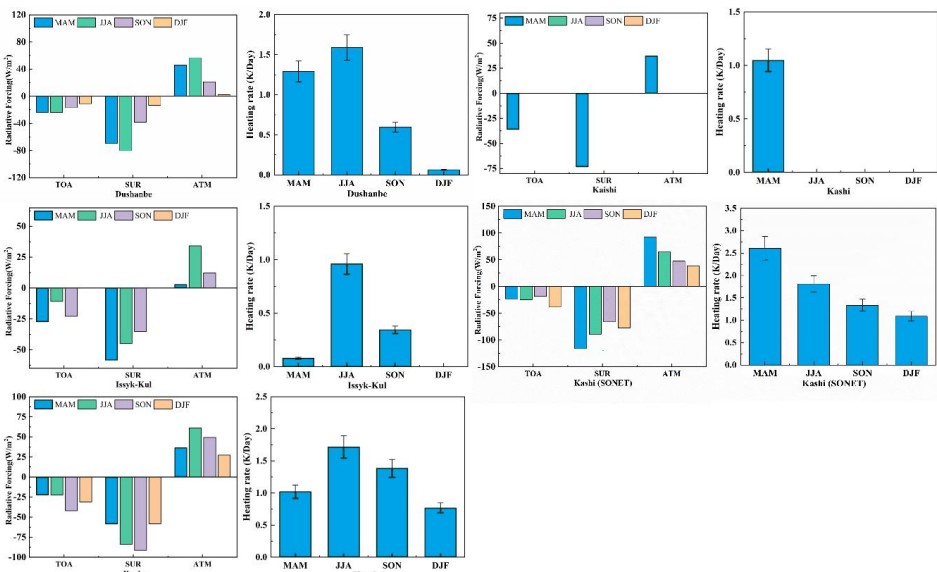

**Figure. 8 Seasonally averaged shortwave radiative forcing and atmospheric heating rate (including direct**
**radiative forcing at the top of the atmosphere (TOA), the surface (SUR), and the atmosphere (ATM)) for dust**
**aerosols at stations in Central Asia.**
Figure 9 provides further refinement of the direct radiative forcing of aerosols at the stations, and the
daily variation of the ADRF demonstrates that the radiative forcing at the top of the atmosphere (TOA),
at the surface (SFC), and in the entire atmosphere exhibits a clear temporal divergence pattern. The ADRF
time series at each site demonstrates a differentiated response: Dushanbe (2011-2023) displays
characteristics typical of inland Central Asia, with the radiative forcing at the TOA and SFC oscillating
within the interval of ±200 W/m² and the atmospheric heating rate peaking at 8 K/day. The short-term
variations are primarily controlled by intermittent dust transport induced by disturbances in the westerly
jet. The Jinghe site demonstrates a generally stable trend, with transient, strong negative forcing (SFC <
-400 W/m²) occurring during extreme dust events. Kashgar, on the other hand, demonstrates significant





temporal variability (TOA/SFC forcing ranging from ±400 W/m² and heating rates from 0-8 K/day
during 2016-2022), particularly high-frequency oscillations during the spring and summer afternoons,
which are directly linked to the "afternoon mixed layer development-dust vertical uplift" mechanism
unique to the Tarim Basin (Nakamae and Takemi 2022).
Notably, the irregular fluctuations in the enhancement of ADRF observed in recent years (2020-2023)
can be attributed to the synergistic effect of changes in surface cover and the frequency of extreme
weather events in the arid zones of Central Asia. Specifically, the significant day-to-day variability
(ΔADRF > 50 W/m²) at the Kashgar site demonstrates the sensitive feedback of aerosol loading on
boundary layer thermal processes in the source region of the Taklamakan Desert. These refined
observations provide critical process evidence for elucidating the transient effects of dust radiative effects
on the regional energy balance.

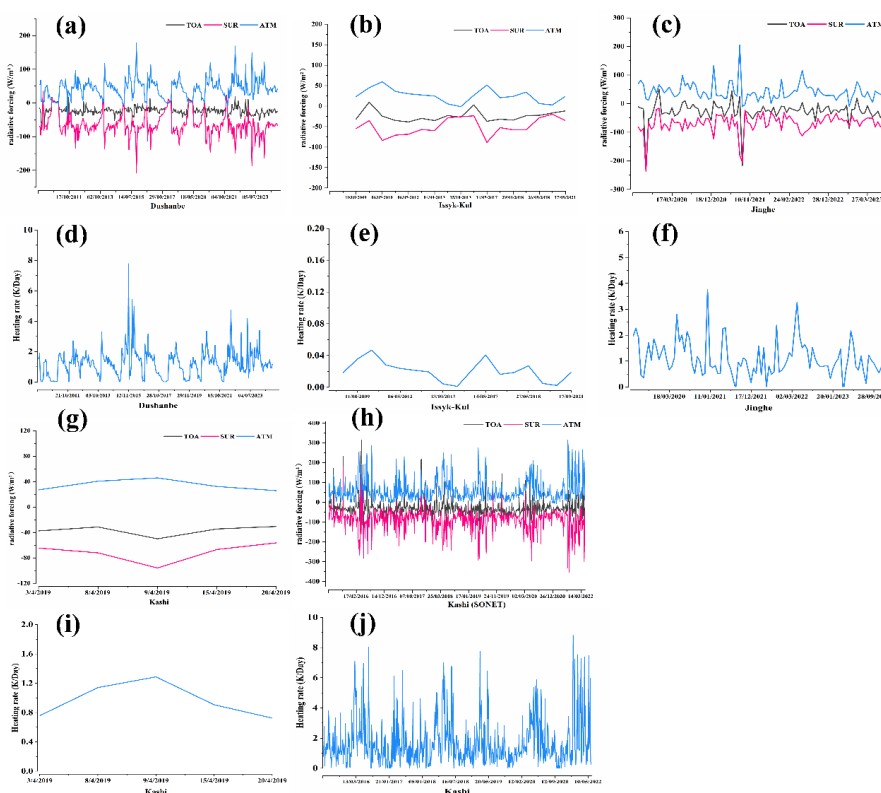

**Figure. 9 Short-wave radiative forcing and atmospheric heating rates (including direct radiative forcing and**
**heating rates at the top of the atmosphere (TOA), surface (SUR), and atmosphere (ATM)) for daily dust**
**aerosols at stations in Central Asia.**



**4. Conclusion and discussion**

**4.1 Conclusion**

Dust aerosols are a key factor in the climate system and are characterized by both complexity and regional variations. This paper compares the spatial distribution and temporal trends, forecasting the future trends of the two data sets in Central Asia based on MERRA-2 observational data and the dust balance data generated by the Multi-model Ensemble of CMIP6 (MME). The comparative analysis from 1980 to 2014 indicates that the reanalyzed data are consistent with the results of the MME simulations. The primary areas of dust emissions include the Tarim Basin, the Aral Sea dry zone, and the Gobi Desert, with maximum emission fluxes greater than 15 $\mu g \cdot m^{-2} \cdot s^{-1}$. Dust emissions in the Aral Sea region increased substantially (>0.5) over the 34 years. In contrast, the emission fluxes in the Tarim Basin increased at an approximate rate of -0.3 $\mu g \cdot m^{-2} \cdot s^{-1} \cdot yr^{-1}$, with a declining trend.

Regarding short-, medium-, and long-term projections, the area of high dust values in Central Asia remains stable in the hinterland of the Aral Sea, Turkmenistan, and along the eastern edge of the Tarim Basin. Short-term emissions in the Aral Sea region range from 17.8 to 26.0 $\mu g \cdot m^{-2} \cdot s^{-1}$, showing minimal variation; however, long-term dust emissions in Central Asia under high radiative forcing scenarios (e.g., SSP585) increase to 387.1 $\mu g \cdot m^{-2} \cdot s^{-1}$, representing an increase of up to 94.9% compared to the reference period. Long-term emissions in the Tarim Basin, on the other hand, demonstrate a declining trend, ranging from 18.7% (SSP245 scenario) to 29.3% (SSP370 scenario), particularly in the SSP585 scenario, with short-term emissions at 27.2 $\mu g \cdot m^{-2} \cdot s^{-1}$, which then decrease to 20.1 $\mu g \cdot m^{-2} \cdot s^{-1}$ in the long term, representing a decrease of 26.1%.

The area of high dust deposition values (>5 $\mu g \cdot m^{-2} \cdot s^{-1}$) overlaps significantly with the emission hotspots. The trend analysis shows that the Aral Sea and the eastern edge of the Caspian Sea exhibit the strongest positive trend ($\Delta S$ = +0.15 $\mu g \cdot m^{-2} \cdot s^{-1}$), while the southern border experiences a negative trend ($\Delta S$ = -0.10 $\mu g \cdot m^{-2} \cdot s^{-1}$). Under the four future scenarios, the influence of dust deposition spreads across southwestern Central Asia, the southeastern edge of the Tarim Basin, and the Junggar Basin, with the maximum deposition flux exceeding 8 $\mu g \cdot m^{-2} \cdot s^{-1}$. The mean values in the near term (2021-2040) range from 9.3 $\mu g \cdot m^{-2} \cdot s^{-1}$ (SSP585) to 10.4 $\mu g \cdot m^{-2} \cdot s^{-1}$ (SSP245), and from 9.6 $\mu g \cdot m^{-2} \cdot s^{-1}$ (SSP370) to 10.0 $\mu g \cdot m^{-2} \cdot s^{-1}$ (SSP126) in the long term (2081-2100), with an overall variation of less than 12%.

The ADRF of dust aerosols in the clear skies of Central Asia exhibits notable spatial patterns. The TOA



radiative forcing is negative overall, with the lowest value observed in the region occurring in the Caspian Sea (<-10 W/m²), followed by the Tarim Basin and around the Aral Sea (<-8 W/m²), with the seasonal minimum in spring (-3.32 W/m²) > summer (-3.21 W/m²) > fall (-3.07 W/m²) > winter (-1.94 W/m²). The SFC reaches its peak at -20 W/m² in the Tarim Basin as well as in southwestern Central Asia. The peak of SFC shortwave radiative losses in the Tarim Basin and southwestern Central Asia reaches a maximum of -20 W/m². The atmospheric shortwave radiative forcing aligns with TOA and SFC in spatial distribution, with a maximum value of 10.02 W/m² in spring, which closely correlates with the seasonal characteristics of dust activities.

The direct aerosol radiative forcing at the sites simulated by the SBDART model was calculated, peaking at the Dushanbe, Issyk-Kul, and Jinghe sites in summer (56.72, 34.22, and 61.17 W/m²), and then dropping to an annual minimum in winter (about 2.33 W/m² in Dushanbe and 27.36 W/m² in Jinghe, respectively), with a peak observed at the Kashgar site in spring (92.99 W/m²). Atmospheric heating rate variations exhibited a strong positive correlation with ADRF. The peaks of heating rates at all sites were observed during the active dust season: Dushanbe (1.29 K/day in summer), Jinghe (1.72 K/day in summer), and Kashgar exhibited a peak in spring (2.61 K/day). Notably, the heating rate at Issyk-Kul was significantly lower in spring (0.08 K/day) than in fall (0.34 K/day), indicating seasonal modulation of dust emission intensity and the influence of local boundary layer processes.

**4.2 Discussion**

In this study, a coupled "emission-deposition-radiation" pathway model was developed for the dust cycle in Central Asia by integrating the MERRA-2 reanalysis data, CMIP6 Multi-Model Ensemble (MME), and ground-based solar photometer observations, with the aim of revealing the radiative forcing mechanism of dust aerosols on the geosphere and the atmosphere. Given the spatial and temporal heterogeneity of aerosol radiative forcing and the scarcity of observational data, a seasonal autoregressive integrated sliding average model (SARIMA) was introduced to forecast and model regional dust radiative forcing using monthly resolution data from 1980 to 2023.



The prediction results (Fig. 10) show that the overall radiative forcing of dust in the Central Asian dry
zone from 2024 to 2029 remains quasi-steady, with interannual fluctuations of ADRF in the range of 1.6-
9.8 W/m², peaking in 2026, and with no extreme event signals detected. The regional differentiation is
characterized by significant features: the southern border, as a substantial radiological response area, is
predicted to reach 2.8-18.9 W/m², while the northern border shows a non-stationary trend (1.6-10.0
W/m²), increasing and then decreasing, possibly due to the bidirectional modulation of dust emissions
by changes in snow cover in the region. Model validation (Supplementary Figure 7) indicated that the
residual series of SARIMA(1,1,0)X(1,0,2)$_{12}$ satisfied the white noise assumption (Ljung-Box Q-test p >
0.05), the probability distributions conformed to N(0,1) normality (K-S test D = 0.12), and the
autocorrelation coefficients of the ACFs lay within the 95% confidence intervals, confirming the model's
predictive reliability.

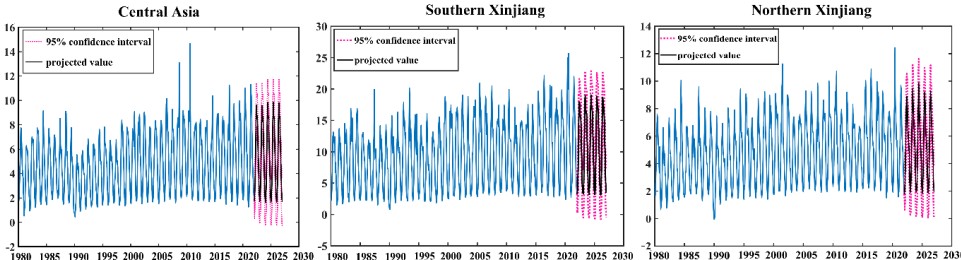

**Figure.10  Dust and sand aerosol direct radiative forcing SARIMA model predictions.**
As a key framework for describing the entire life cycle of the dust cycle, the dust balance encompasses
critical aspects such as emission, dispersion, deposition, mass loading, lifetime, and optical depth (DOD).
Despite progress in this field, a deeper understanding of the complex interactions between dust with land,
vegetation, and climate still faces significant challenges. In particular, the diversity in assumptions
regarding dust particle size in CMIP6 models significantly affects the consistency of simulation results,
thereby increasing the uncertainty in simulating dust cycling processes.
Specifically, the influence of the spatial and temporal variability of the dust particle size spectrum on
radiative forcing has often been neglected, particularly the rapid settling of coarse particles (>10 μm),
which may result in an underestimation of the longwave radiative effect of dust transported over long
distances. Therefore, an in-depth investigation of the diversity in the particle size spectrum and its
mechanisms on the radiative effect of dust is crucial for understanding regional differences in the dust
cycle and their radiative impacts, particularly in the context of extreme climatic events, such as





sandstorms and dust storms.
In addition, the current uncertainty in the simulation of dust aerosols in CMIP6 models not only persists
but is also increasing (Wang et al. 2021), which makes it more difficult to understand the full life cycle
of dust and its interactions with other components of the climate system (e.g., radiation budget, cloud
processes, precipitation, and atmospheric circulation). This reflects the lack of practical constraints on
the links between dust processes, highlighting the urgent need to develop more accurate models of dust
aerosol processes to enhance the precision of climate predictions.
It is worth noting that existing radiation transfer models, such as SBDART, while valuable in their
applications, fail to fully incorporate aerosol-cloud interactions, which are particularly important in
regions with high dust concentrations. Aerosol-cloud interactions significantly affect radiative forcing,
and neglecting this factor may lead to discrepancies in radiative forcing estimates. Therefore, future
research should integrate high-resolution models that account for aerosol-cloud interactions and refine
the vertical distribution and particle size spectrum characteristics of dust to improve the accuracy of
radiative forcing calculations and enhance scientific understanding of the climatic effects of dust.
**Author contributions**
All authors contributed to the manuscript and approved the final version. YG designed the study,
performed the data analysis, and wrote the original draft. WC, JD, and YR assisted with data collection
and software processing. YR also contributed to the validation and interpretation of results. ZZ
supervised the research and contributed to manuscript revision and funding acquisition.
**Competing interests.**
The contact author has declared that none of the authors has any competing interests.
**Funding**
National Natural Science Foundation of China (No. 42061066), and Open Project of Key Laboratory in
Xinjiang Uygur Autonomous Region of China (2023D04066)



**Acknowledgements**
We are sincerely grateful to MERRA-2 (https://gmao.gsfc.nasa.gov/reanalysis/MERRA-2/), CMIP6
(https://esgf-node.ipsl.upmc.fr/projects/cmip6-ipsl/), AERONET and SONET Instrument Science
Teams for providing the datasets, which are available on the Web and constitute the centralized
database for the current work. We thank the faculty members who worked hard to review this thesis.

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
