# Peer review of "Assessment and prediction of dust emissions, deposition"

_EGUsphere, 2025_

## Author Comment (AC1)

Thank you for your decision and constructive comments on my manuscript. We have carefully considered the reviewers' suggestions and made corresponding changes. We have tried our best to improve the manuscript accordingly.

In accordance with ACP discussion policies, the full revised manuscript paragraphs are not displayed in these responses. Instead, for each comment, we provide a concise summary of the modifications made and indicate the specific sections, page numbers, and line numbers in the revised manuscript where the changes can be found. We hope that this format allows the reviewers to efficiently assess the revisions while maintaining clarity and compliance with the journal's guidelines.

The parts highlighted in red in the revised manuscript have been updated according to your comments. Revision notes, point-by-point, are given as follows:

**RC1:**

**Major comments:**

(1) Some figures are difficult to interpret without clear legends and labels. For instance the unit in figure 6 caption:  $\mu$ g-m-2-s-1, the superscripts should be typeset properly. I recommend change the font size for figures.

We sincerely thank the reviewer for the valuable comments. Due to an oversight, some figure legends, labels, and unit formatting were not sufficiently clear. In response, the entire manuscript—including all figures—has been carefully reviewed. Figure 6, in particular, has been completely redrawn and reformatted, with legends and labels revised or supplemented where necessary. Additionally, the formatting of units, including superscripts and subscripts, has been standardized throughout. We greatly appreciate your careful review and constructive suggestions.( manuscript pages 17, lines 380)

**Revised Figure:**

Figure. 6 Time evolution of the dust budget. Dust emissions, dry and wet deposition ( $\mu g \cdot m^{-2} \cdot s^{-1}$ ) for panels (a–c) northern Xinjiang, (d–f) southern Xinjiang, (g–i) Central Asia, and (j–l) the entire study region. Results are from the CMIP6 multi-model ensemble (MME; 1980 – 2100) and MERRA-2 (1980 – 2023).

**(2) Please proofread the whole manuscript. There are occasional grammar issues and awkward phrasing. Most of them do not substantially impede understanding, though some do affect the clarity.**

We sincerely thank the reviewer for the careful comments. In response to the identified grammatical issues and occasional awkward phrasing, the entire manuscript has been thoroughly reviewed and revised on a sentence-by-sentence basis. Particular attention was given to correcting grammatical errors and improving unclear or awkward expressions, ensuring overall clarity, accuracy, and readability.

(3) This study uses a combination of MERRA-2, CMIP6, and SBDART, but it is not very clear what is genuinely new compared to previous dust studies over Central Asia. The authors should clarify what exactly is new in this work and how this advances beyond previous studies.

We sincerely thank the reviewer for this insightful comment. In the revised

manuscript, we have clarified the novelty and advancement of our study in the Introduction section (see revised version, pp. 3-4, lines 65-104, 1. Introduction). The main innovations and contributions of this work are summarized as follows:

Previous pioneering studies (e.g., Kok et al., 2023; Zhao et al., 2022 - 2024) have greatly enhanced our understanding of the global dust cycle, but their emphasis has been primarily global in scope, where coarse spatial resolution constrains the characterization of regional processes over complex terrains such as Central Asia. Our study introduces a regional-scale multi-source data fusion framework that systematically integrates the full "emission - deposition - radiative forcing" chain. This framework enables a more refined depiction of the dust life cycle in arid Central Asia, effectively bridging the gap between global simulations and regional processes.

Beyond previous regional studies that primarily examined dust distribution and transport (e.g., Li et al., 2022b; Tao et al., 2022), this work advances the field in three key aspects:

- (1) By integrating MERRA-2 reanalysis with the CMIP6 multi-model ensemble and applying a statistical downscaling approach, we quantified the long-term evolution and scenario-dependent divergence of dust emission deposition budgets in Central Asia.
- (2) Using the SBDART radiative transfer model constrained by ground-based observations from the SONET network and the Jinghe CE318 site, we obtained refined estimates of clear-sky shortwave dust radiative forcing.
- (3) A SARIMA statistical model was further employed to examine short-term variations and potential risks of dust radiation interactions, offering a novel perspective for near-term environmental prediction.

Importantly, the study also emphasizes reducing model uncertainty through observational constraints, thereby bolstering the credibility of radiative forcing assessments in data-scarce regions. Overall, this work delivers one of the first integrated assessments linking dust emission, deposition, and radiative effects over the complex terrain of Central Asia. The proposed framework not only deepens our understanding of regional dust – climate interactions but also provides scientific support for enhancing climate model performance and environmental risk management in arid regions.

**Revised excerpt (pp. 3-4, lines 85-95):**

"To overcome the above limitations, this study establishes a multi-source data integration framework that systematically incorporates a full-chain analysis of 'emission-deposition-radiation,' focusing on the regional characteristics of the dust lifecycle over

the arid regions of Central Asia. Unlike previous studies that have mainly concentrated on the global scale (Kok et al., 2023; Zhao et al., 2022, 2023, 2024), this work achieves an integrated assessment at the regional scale and emphasizes the role of observations in constraining model uncertainties. The specific innovations are reflected in three aspects: (1) by combining MERRA-2 reanalysis with CMIP6 multi-model ensembles through statistical downscaling... [further details on (2) and (3) as summarized above]."

(4) The results show strong vertical gradients in radiative forcing and surface cooling. However, the physical interpretation in the manuscript is limited. For instance, what are the implications for regional climate? More discussion regarding the broader implication would be beneficial.

We sincerely thank the reviewer for this valuable comment, which highlights a crucial aspect of our study. A thorough elucidation of the physical mechanisms underlying the vertical gradient of radiative forcing and its climatic implications substantially enhances the scientific value of the research. Accordingly, the manuscript has been substantially revised and supplemented (primarily on pp. 17–18, lines 393–416, 3.3.1 Monthly average changes in direct radiative forcing by dust aerosols) to strengthen the physical interpretation and expand the discussion of its regional climate significance.

The specific revisions and supplements are summarized as follows:

Enhanced explanation of the "surface cooling-atmospheric heating" mechanism: The results indicate that the pronounced negative surface radiative forcing (with cooling centers reaching –20 W/m²) primarily arises from the combined effects of dust scattering and absorption of solar radiation, significantly reducing net surface radiation. The resultant surface energy deficit further weakens sensible heat flux and evaporation, thereby suppressing the exchange of heat and moisture from the surface to the atmosphere. This provides a clear physical depiction of how dust alters the surface—atmosphere energy balance to establish a vertical radiative gradient.

**In-depth discussion of potential regional climate impacts and significance:**

Impact on aridity and the water cycle: Surface cooling and reduced evapotranspiration may exacerbate inherent water deficits in the arid regions of Central Asia, posing potential constraints on vegetation growth and agricultural water resources.

Feedback on dust activity: Atmospheric heating enhances the vertical temperature gradient, which may increase the likelihood of deep convection, thereby potentially amplifying the frequency and intensity of spring-summer dust storms. This "dustradiation-dynamics" feedback loop represents a key finding of this study.

Implications for weather and ecosystems: Based on observed strong day-to-day fluctuations in ADRF at the measurement sites, such instantaneous radiative perturbations may alter boundary layer stability and convective potential, consequently affecting regional precipitation variability and ecosystem stability.

**Revised excerpt (from pp. 17-18, lines 393-416):**

"The spatial pattern of surface (SFC) radiative forcing (Figure 7e-h) exhibits stronger negative values, with two pronounced cooling centers over the Tarim Basin and southwestern Central Asia, where shortwave radiation loss peaks at -20 W/m2. This arises from the combined scattering and absorption effects of dust particles on incoming solar radiation (Li et al., 2022a), which substantially reduce surface net radiation, thereby diminishing sensible heat flux and evaporation processes and suppressing the transfer of heat and water vapor from the surface to the atmosphere. The atmospheric radiative forcing (ADRF) exhibits a spatial pattern consistent with those at the TOA and SFC but features positive values (10.02 W/m2 in spring and 9.89 W/m2 in summer), indicating the energy redistribution role of dust aerosols in trapping solar energy within the atmospheric system via shortwave absorption. This vertical gradient of 'surface cooling and atmospheric heating' induces substantial changes in the regional thermodynamic structure (Kok et al., 2017). On one hand, surface cooling diminishes sensible heat flux and evaporation, thereby exacerbating moisture deficits in Central Asia's arid regions and limiting vegetation growth and agricultural productivity. On the other hand, atmospheric heating strengthens the temperature gradient from the boundary layer to the free troposphere, enhancing the potential for deep convection, which could intensify the frequency and severity of spring-summer dust storms and modify regional precipitation patterns and extreme weather events."

**Technical comments:**

**(1) Please define all the acronyms at their first use.**

We sincerely thank the reviewer for the careful reminder. The manuscript has been thoroughly checked to ensure that all abbreviations are fully defined upon their first appearance. The main abbreviations include:

**MERRA-2:** Modern-Era Retrospective analysis for Research and Applications, Version 2

**CMIP6:** Coupled Model Intercomparison Project Phase 6

SBDART: Santa Barbara DISORT Atmospheric Radiative Transfer model

ADRF: Aerosol Direct Radiative Forcing

TOA: Top of Atmosphere

SFC: Surface

**SSPs:** Shared Socioeconomic Pathways

(2) The graphical abstract is overloaded with text and figures, which makes it difficult to read the key message and results in low readability. The author is encouraged to simplify the graphical abstract by focusing on the main story and remove secondary details.

We sincerely appreciate the reviewer's constructive comment. In response, the graphical abstract has been completely redesigned. The revised version removes redundant text and secondary graphical details, focusing instead on the core research storyline—from dust emission and deposition to radiative forcing and climate feedback. This streamlined design enhances visual clarity, emphasizes the key information, and substantially improves readability.

**Revised Figure:**

(3) The linear trend analysis was used throughout the manuscript, but it is not clear whether statistical significance was tested. Please clarify the method used to calculate the trends and whether the reported values are statistically significant.

We sincerely thank the reviewer for this important comment, which underscores a critical aspect of our analysis. Evaluating the statistical significance of trends is indeed essential for bolstering the reliability of the study's conclusions.

To address this, we have incorporated detailed information on the trend analysis and significance testing in Section 2.2.4 ("Linear Trend Method"; revised manuscript, p. 10, lines 240–247). Specifically:

Trend calculation method: Linear trends were estimated using ordinary least squares (OLS) regression to derive the slope.

Statistical significance testing: All trends were assessed via a two-tailed t-test, with p < 0.05 deemed statistically significant.

Visualization of significance: In the spatial trend maps, grid points exhibiting significant trends (p < 0.05) are denoted by dots.

Regional average trends: In the time series plots of regional mean trends, both the regression slope and the corresponding p-value are explicitly annotated on the figures.

**Revised excerpt (from p. 10, lines 240-247):**

**2.2.4 Linear Trend Method**

In this study, the ordinary least squares (OLS) method was used to perform linear regression on the dust budget time series, with trend significance assessed via a two-tailed t-test. Spatial trends were derived by conducting independent regressions at each grid point, with statistically significant results (p < 0.05) indicated by stippling in the figures. Regional mean trends were calculated by regressing the annual averages of grid values within specific regions (Central Asian countries, northern Xinjiang, and southern Xinjiang). The regression slopes and corresponding p-values were annotated directly on the time series plots.

**(4) The description of the downscaling method lacks clarity on uncertainties. Please provide more justification for the choice of this method versus dynamic downscaling and quantify potential biases introduced.**

We sincerely thank the reviewer for this insightful comment, which has helped substantially enhance the methodological rigor and deepen the discussion of uncertainties. In response, extensive revisions and additions have been made to the manuscript, primarily in Section 4.2.1 ( "Uncertainty Analysis"; revised manuscript, pp. 25 - 28, lines 570 - 635) and the associated methodological descriptions. The specific clarifications are summarized as follows:

**Rationale for method selection:**

The reasons for adopting statistical downscaling (the Delta method) have been clarified as follows:

Computational efficiency: Dynamic downscaling demands substantial computational resources, whereas the Delta method facilitates efficient analyses across long-term, multi-model, and multi-scenario simulations.

Reduced sensitivity to parameterization uncertainty: Dynamic downscaling outcomes are highly sensitive to internal parameterizations, which can introduce additional, difficult-to-quantify biases in observation-sparse regions such as Central Asia.

Preservation of statistical relationships: The Delta statistical downscaling method effectively accommodates Central Asia 's complex terrain while maintaining the statistical relationships between dust emissions and climate variables, rendering it well-suited for this study.

**Quantification of potential biases:**

Two primary sources of uncertainty have been systematically evaluated and quantified:

Inter-model uncertainty: Figure 11a and Supplementary Figure 12 spatially illustrate the biases of individual models in the multi-model ensemble (MME) relative to MERRA-2 reanalysis data. Systematic positive/negative biases are evident in key source regions, such as the Taklamakan and Karakum deserts. The inherent dispersion among model simulations is further visualized via the  $\pm 1\,\sigma$  standard deviation of the MME (shaded area in Figure 10b).

**Bias introduced by the downscaling procedure:**

The performance of the statistical downscaling was quantified using spatial root-mean-square error (RMSE) to identify areas of higher uncertainty (e.g., RMSE > 2  $\mu$ g m-2 s-1 in complex terrains such as the Tarim Basin; Figure 11b).

Scatterplots (Supplementary Figure 13a) reveal an overall bias of -1.26 and RMSE of 4.31, with a coefficient of determination ( $R^2 > 0.91$ ), demonstrating high fidelity in capturing spatial patterns despite minor systematic underestimation.

Time series comparisons (Supplementary Figure 13b) confirm that the method reasonably reproduces observed interannual and seasonal variability.

**Revised excerpt (from pp. 58-68, lines 570-635, 4.2.1 Uncertainty Analysis):**

"The bias-corrected statistical downscaling method employed in this study, which relies on MERRA-2 data, is well-suited to Central Asia's complex terrain and sparse observational networks. It offers low computational costs while preserving the statistical

relationships between dust emissions and climate variables—advantages over dynamic downscaling, which is resource-intensive and sensitive to parameterization uncertainties. However, its capacity to simulate extreme events (e.g., intense dust storms) remains limited. To quantify downscaling biases, Figure 11b illustrates the spatial RMSE between CMIP6 downscaled outputs and MERRA-2 data (RMSE > 2  $\mu$ g·m-2·s-1 in the Tarim Basin and Karakum Desert). Supplementary Figure 13a shows a high correlation (R² > 0.91) with minor underestimation (Bias = -1.26, RMSE = 4.31), and time series (Figure 13b)

confirm reproduction of variability."

Figure. 11 (a) Dust emission biases of individual models relative to MERRA-2 reanalysis data; (b) time series of dust emissions from the CMIP6 multi-model ensemble, with  $\pm 1\sigma$  inter-model variability indicated by the shaded area.

Figure. 12 (a) Spatial distribution of biases between individual models in the multi-model ensemble and MERRA-2 reanalysis data; (b) spatial distribution of root-mean-square error (RMSE) between CMIP6 downscaled outputs and MERRA-2 reanalysis data.

(5) For the SBDART simulations, it is not clear how representative the ground-based input data are across such a vast heterogeneous region. How many stations, and how are they weighted? More detail on spatial representativeness and limitations would strengthen the analysis.

We sincerely thank the reviewer for raising this critical point. We fully agree that clarifying the representativeness of ground-based observations is essential for correctly

interpreting the SBDART simulation results. In response, we have added a detailed explanation and clarification in the revised manuscript (p. 19, lines 421 - 434, Section 3.3.2 "Refinement of Aerosol Direct Radiative Forcing in Dusty Weather"). The revisions are summarized as follows:

**(1) Number, location, and representativeness of observation sites**

The SBDART simulations in this study were based on long-term aerosol optical properties observed at five ground-based sites, including:

Dushanbe (AERONET): representing the inland region of Central Asia;

Issyk-Kul (AERONET): representing high-altitude lake environments;

Kashgar (AERONET and SONET): representing the dust source region of the Tarim Basin;

Jinghe (SONET): representing the Gobi - desert transition zone.

These sites encompass the main geomorphological units and typical surface types of Central Asia, enabling spatial representation of the regional diversity in dust aerosol radiative effects. Although the number of sites is limited, their strategic spatial distribution ensures representativeness. Thus, our analysis emphasizes localized radiative responses under distinct environmental conditions rather than spatially averaged regional means.

**(2) Weighting strategy and spatial representativeness**

No spatial weighting was applied among these sites, primarily for the following reasons:

Long-term, continuous aerosol observations are extremely scarce across Central Asia, and these five sites constitute the full set of available high-quality datasets.

Applying spatial weighting with limited and unevenly distributed sites may introduce misleading regional biases. Therefore, each site was treated as an independent case study to highlight differences and similarities in dust radiative forcing under varying terrain and climatic settings.

**(3) Discussion on spatial representativeness and limitations**

A new paragraph has been added in the revised manuscript to explicitly address these limitations:

Simulations based on limited sites can only reflect localized radiative responses in representative regions, rather than capturing the full spatial heterogeneity of Central Asia.

As exemplified by Issyk-Kul, high-altitude topography and its shielding effects may weaken springtime dust vertical transport, potentially leading to underestimation of local

radiative forcing and underscoring challenges in mountain observations.

Moreover, cross-validation with MERRA-2 reanalysis data (Supplementary Fig. S1) confirms the overall consistency and reliability of the site observations at a regional scale.

Finally, the Conclusion section now explicitly recommends that future research enhance spatial representativeness by expanding ground-based networks and integrating satellite – model synergistic retrievals.

**Revised excerpt (from p. 19, lines 421–434):**

"These simulations are based on ground-based observations from 2011 to 2023, encompassing AERONET sites at Dushanbe (Tajikistan; representing the Central Asian interior), Issyk-Kul (Kyrgyzstan; representing high-altitude lake regions), and Kashgar (Xinjiang, China; representing the Tarim Basin dust source region), as well as the SONET site at Kashgar and our self-established Jinghe site (Xinjiang, China; representing the Gobi-desert transitional zone). Although the number of sites is limited, their spatial distribution covers the primary dust source regions and representative surface types, thereby achieving a degree of regional representativeness. No spatial weighting was applied, as long-term observations are scarce in Central Asia, and treating sites as independent case studies avoids introducing biases from uneven distribution. Limitations include the inability to fully capture spatial heterogeneity; for instance, Issyk-Kul's topography may underestimate dust transport. Cross-validation with MERRA-2 (Supplementary Fig. S1) confirms site reliability at the regional scale."

(6) The SARIMA model is interesting but feels a bit disconnected, it is not well integrated with the rest of the results. The justification for why a short-term statistical forecast is useful in this context should be expanded. Also please add error metrics to evaluate this forecast skill quantitatively.

We sincerely thank the reviewer for highlighting this important point regarding the integration of the SARIMA model within the overall research framework and its predictive relevance. This comment has greatly assisted us in refining the logical structure and enhancing the scientific rigor of our analysis. In response, we have substantially revised the manuscript to clarify the methodological role, integration, and quantitative validation of the SARIMA model. The main revisions are summarized as follows (pp. 24–25, lines 536–568, Section 4.2.1 "SARIMA Forecasting"):

(1) Scientific positioning of the SARIMA model and its integration into the overall framework

The revised manuscript clearly positions the SARIMA model as a methodological supplement rather than an independent analytical module. It is designed to bridge the temporal gap between the historical diagnostic analysis based on MERRA-2 reanalysis and SBDART ground-based simulations, and the long-term projections from CMIP6 scenario experiments. While CMIP6 primarily captures centennial-scale trends, SARIMA effectively characterizes interannual and short-term internal variability in dust radiative forcing, thereby linking long-term climate projections with near-future variations. Consequently, SARIMA serves as a key component of the integrated "emission—deposition—radiation" framework, enabling operational prediction and temporal continuity within the system.

**(2) Scientific rationale and practical significance of short-term statistical prediction**

In response to the reviewer's concern about the necessity of short-term forecasting, the revised text elaborates further on its scientific justification and real-world relevance. Dust radiative forcing exhibits strong spatiotemporal non-stationarity and high uncertainty, compounded by limited observational data over Central Asia. Under these conditions, short-term statistical prediction provides actionable insights that complement long-term climate projections. The SARIMA model identifies potential dust-active periods within the next 3–5 years and quantifies the magnitude of interannual fluctuations. These results offer scientific support for regional assessments of agricultural productivity, water resources, renewable energy planning, and ecological risk management, thereby enhancing the operational applicability of the study.

**(3) Quantitative validation of model performance**

Following the reviewer's suggestion, we have added quantitative validation metrics for the SARIMA model's predictive performance (Section 4.2.1 and Supplementary Figure 11). The results demonstrate RMSE = 1.72 W/m², MAE = 1.21 W/m², MAPE = 8.6%, and R² = 0.70, indicating high predictive accuracy in capturing short-term variability. Furthermore, the residual series passed both the Ljung - Box Q test (p > 0.05) and the Kolmogorov - Smirnov normality test (D = 0.12), confirming the model's statistical robustness.

Through these revisions, the functional role of the SARIMA model is now clearly delineated. It complements CMIP6-based long-term simulations in the temporal dimension, while its quantitative validation enhances both the scientific reliability and practical value of the results. By integrating SARIMA into the analysis framework, the

study provides a more complete representation of dust-climate interactions and offers actionable guidance for regional climate risk management and near-future adaptation planning.

**Revised excerpt (from pp. 24 - 25, lines 536 - 569):**

"This study integrates MERRA-2 reanalysis data, CMIP6 multi-model ensemble (MME) simulations, and ground-based sun photometer observations to develop a fully coupled 'emission - deposition - radiation' framework for the dust cycle in Central Asia, thereby systematically elucidating the radiative regulatory mechanisms of dust aerosols on the land - atmosphere system. To address the pronounced spatiotemporal heterogeneity in aerosol radiative forcing and the limitations of observational data, this study employs a seasonal autoregressive integrated moving average (SARIMA) model as a methodological supplement, bridging the temporal gap between historical diagnostics (MERRA-2/SBDART) and long-term CMIP6 projections by capturing interannual and short-term variability. In contrast to century-scale CMIP6 simulations, SARIMA quantifies short-term internal variability (Kumar et al., 2018), providing actionable near-term forecasts (3 - 5 years) for dust-active periods, which support regional risk assessments in agriculture, water resources, and ecology. Model validation (Supplementary Figure 11) shows RMSE = 1.72 W/m2, MAE = 1.21 W/m2, MAPE = 8.6%, R2 = 0.70; residuals pass Ljung - Box Q test (p > 0.05) and Kolmogorov - Smirnov test (D = 0.12), confirming robustness. Thus, SARIMA complements the framework, enhancing operational applicability for near-future adaptation."

We sincerely appreciate your guidance and suggestions. We will continue to work diligently to improve and refine our research. Thank you, reviewers and editors, for your enthusiastic contributions. We hope the corrections will be approved. In line with ACP guidelines, the full revised manuscript will be submitted separately upon completion of the discussion phase.

---

## Author Comment (AC2)

Thank you for your decision and constructive comments on my manuscript. We have carefully considered the reviewers' suggestions and made corresponding changes. We have tried our best to improve the manuscript accordingly.

In accordance with ACP discussion policies, the full revised manuscript paragraphs are not displayed in these responses. Instead, for each comment, we provide a concise summary of the modifications made and indicate the specific sections, page numbers, and line numbers in the revised manuscript where the changes can be found. We hope that this format allows the reviewers to efficiently assess the revisions while maintaining clarity and compliance with the journal's guidelines.

The parts highlighted in red in the revised manuscript have been updated according to your comments. Revision notes, point-by-point, are given as follows:

**RC2:**

**Major comments:**

(1) The claim of presenting the first "closed-loop emission-deposition-radiation analysis" should be moderated. Several previous studies (e.g., Kok et al., 2023; Zhao et al., 2022, 2023, 2024) have examined the dust life cycle, particularly at the global scale. While the proposed "closed-loop" framework is conceptually interesting, its interpretation and scientific depth remain somewhat superficial. A more detailed discussion of how this framework advances understanding beyond prior studies would strengthen the contribution of the manuscript.

We sincerely thank the reviewer for this constructive comment. The concern regarding the originality and scientific depth of the "closed-loop emission–deposition–radiation analysis" framework is highly valuable for refining the scientific positioning and rigor of our study. In response, we have carefully revised the manuscript to systematically clarify the research scope, its relationship to previous studies, and the innovative aspects of the framework. The main revisions are summarized as follows: (pp. 3–4, lines 65–104, Section 1. "1.introduction"):

**(1) Removal of overemphasized claims of novelty**

All statements in the original manuscript referring to the "first" or "first-ever closed-loop analysis" have been completely removed to avoid overstating the innovation. The revised core discussion now focuses on presenting a multi-source data integration framework that systematically quantifies dust emission—deposition—radiation processes at the regional scale, thereby enabling a full-chain diagnosis of the dust lifecycle. This

revision more accurately reflects the study's contribution and positioning.

**(2) Relationship to and distinction from existing global-scale studies**

In the Introduction, we have added a systematic review and clarification regarding previous work. We explicitly state that this study does not replicate global-scale analyses such as those by Kok et al. (2023) or Zhao et al. (2022, 2023, 2024), but rather deepens and extends their findings at the regional scale. While global studies primarily focus on the overall dust budget in the climate system and its impact on radiative balance, our work emphasizes the regional variability of dust emission, deposition, and radiative effects under the complex topography of arid Central Asia. By integrating multi-source observations with model simulations, we reveal region-specific interactions and mechanisms, thereby addressing the limitations of global models in capturing local-scale processes.

**(3) Scientific deepening and innovative contributions of the framework**

To address the reviewer's comments on "scientific depth," the revised manuscript further elaborates the theoretical significance and methodological innovations of the framework in the Discussion:

Regional refinement: By combining CMIP6 global model outputs with MERRA-2 reanalysis and ground-based observations, the framework reveals the long-term evolution and scenario-dependent differences of dust emission—deposition budgets in Central Asia, bridging the scale gap between global models and regional realities.

Observation-constrained modeling: SBDART radiative simulations are constrained using ground-based observations from SONET and AERONET, significantly reducing uncertainty in clear-sky aerosol direct radiative forcing estimates.

Temporal scale complementarity: The incorporation of the SARIMA statistical model captures short-term variability in dust radiative forcing, complementing CMIP6 long-term climate scenarios and providing actionable insights for risk assessment and climate adaptation.

The revised manuscript no longer frames the framework as a "first-of-its-kind" theory, but clearly emphasizes its applied innovation and methodological integration at the regional scale. Through the synergistic fusion of multi-source data and cross-scale analysis, this framework offers new scientific perspectives and empirical support for understanding regional dust lifecycle responses in arid Central Asia, validating climate model simulations, and supporting regional climate risk management.

Revised excerpt (pp. 3-4, lines 85-95):

"To overcome the above limitations, this study establishes a multi-source data integration framework that systematically incorporates a full-chain analysis of 'emission-deposition-radiation,' focusing on the regional characteristics of the dust lifecycle over the arid regions of Central Asia. Unlike previous studies that have mainly concentrated on the global scale (Kok et al., 2023; Zhao et al., 2022, 2023, 2024), this work achieves an integrated assessment at the regional scale and emphasizes the role of observations in constraining model uncertainties. The specific innovations are reflected in three aspects: (1) by combining MERRA-2 reanalysis with CMIP6 multi-model ensembles through statistical downscaling... [further details on (2) and (3) as summarized above]."

(2) The manuscript presents ensemble mean maps and time series for future projections, but the inter-model spread is not shown or sufficiently discussed. Given the known diversity in dust emission schemes and particle size assumptions across CMIP6 models, the robustness of the results cannot be fully assessed from the ensemble mean alone. For example, the large differences in emission trajectories between the Aral Sea and Tarim Basin under SSP585 may depend heavily on the dust emission parameterizations of specific models. To strengthen confidence in the findings, it would be helpful to include an explicit representation of the inter-model spread and a more thorough discussion of how much weight should be placed on the ensemble mean versus the range of individual model outcomes.

We sincerely thank the reviewer for this important comment. As noted, fully assessing inter-model uncertainties is essential for evaluating the robustness of multi-model ensemble (MME) results. Given the substantial differences among CMIP6 models in dust emission schemes, particle sizes, and surface parameterizations, relying solely on ensemble means is insufficient. In response, the revised manuscript adds a new section, "4.2.1 Uncertainty Analysis" (pp. 25–28, lines 571–635), with figures and quantitative results illustrating spatial patterns of inter-model differences and their influence on ensemble mean reliability.

**(1) Visualization of inter-model variability:**

Figures 10a-b quantitatively illustrate model differences and variability. Figure 10a depicts the emission bias of each model relative to MERRA-2 reanalysis, with statistical significance assessed using two-tailed t-tests (p < 0.05). The results reveal significant biases in CESM2, CESM2-WACCM, CNRM-ESM2-1, and MRI-ESM2-1, suggesting that their dust parameterizations may introduce additional uncertainties. Figure 10b presents

time series of dust emissions along with the  $\pm 1\sigma$  inter-model variability range (shaded), providing a quantitative measure of ensemble uncertainty. Notably, MERRA-2 observations consistently fall within the historical model range, demonstrating that the ensemble mean reasonably represents the climatological mean state.

**(2) Regional differences and spatial distribution of model biases:**

Figure 11a and Supplementary Figure 12 reveal the spatial distribution of model biases. All models exhibit biases in key dust source regions, including the Taklamakan Desert, Kumtag Desert, areas surrounding the Aral Sea, Karakum Desert, and the eastern Caspian region. Positive biases predominate in the southeastern Tarim Basin, whereas negative biases dominate the western Karakum and Aral Sea regions. These regional differences confirm the reviewer's concern regarding model dependence, as dust emission trajectories strongly depend on specific emission parameterizations and particle size assumptions.

**(3) Bias correction and downscaling evaluation:**

To further mitigate systematic model errors, a bias-corrected statistical downscaling approach based on MERRA-2 was applied to improve simulations under Central Asia's complex topography. Figure 11b shows the spatial distribution of root-mean-square error (RMSE) between downscaled results and MERRA-2, highlighting elevated RMSE (>2  $\mu$ g m-2 s-1) in topographically complex regions such as the Tarim Basin and Karakum Desert, suggesting caution in interpreting predictions for these areas. Supplementary Figure 13 scatterplots indicate high correlation between downscaled MME anomalies ( $\Delta$ MME) and MERRA-2 (R2 > 0.91), with slight underestimation (Bias = -1.26, RMSE = 4.31). Time series comparisons (Supplementary Figure 11b) also demonstrate that downscaled results effectively capture seasonal and interannual variability.

**(4) Representativeness of ensemble mean and discussion of weighting:**

The revised manuscript further clarifies the physical meaning of the ensemble mean and the rationale for weighting. Despite inter-model variability, the ensemble mean consistently captures the core spatial structure of dust emissions, reflecting the major emission patterns of the "Tarim Basin–Aral Sea–Karakum" source regions. The  $\pm 1\sigma$  variability range and spatial bias maps (Figures 10b and 11a) together constitute an essential component of uncertainty assessment, identifying low-confidence and high-RMSE areas, thereby enhancing the interpretability and practical utility of the results.

**(5) Comprehensive assessment and limitations:**

While the above analyses significantly improve understanding of model distributions

and ensemble robustness, some limitations remain. Differences in dust particle size assumptions and parameterizations remain the primary sources of inter-model variability. Additionally, the SBDART model does not explicitly include aerosol—cloud interactions in radiative transfer calculations, which may bias radiative forcing estimates in high-dust regions. The limited number of ground-based observation sites also constrains representativeness in Central Asia's complex topography. Future studies should integrate higher-resolution models and multi-source observational data, while systematically considering aerosol vertical structure and cloud-phase processes, to further improve the accuracy and reliability of regional radiative forcing assessments.

**Revised excerpt (from pp. 58-68, lines 570-635, 4.2.1 Uncertainty Analysis):**

"The bias-corrected statistical downscaling method employed in this study, which relies on MERRA-2 data, is well-suited to Central Asia's complex terrain and sparse observational networks. It offers low computational costs while preserving the statistical relationships between dust emissions and climate variables—advantages over dynamic downscaling, which is resource-intensive and sensitive to parameterization uncertainties. However, its capacity to simulate extreme events (e.g., intense dust storms) remains limited. To quantify downscaling biases, Figure 11b illustrates the spatial RMSE between CMIP6 downscaled outputs and MERRA-2 data (RMSE > 2  $\mu$ g·m-2·s-1 in the Tarim Basin and Karakum Desert). Supplementary Figure 13a shows a high correlation (R² > 0.91) with minor underestimation (Bias = -1.26, RMSE = 4.31), and time series (Figure 13b) confirm reproduction of variability."

Figure. 11 (a) Dust emission biases of individual models relative to MERRA-2 reanalysis data; (b) time series of dust emissions from the CMIP6 multi-model ensemble, with  $\pm 1\sigma$  inter-model variability indicated by the shaded area.

Figure. 12 (a) Spatial distribution of biases between individual models in the multi-model ensemble and MERRA-2 reanalysis data; (b) spatial distribution of root-mean-square error (RMSE) between CMIP6 downscaled outputs and MERRA-2 reanalysis data.

(3) The SARIMA analysis is an interesting addition, but its role in the manuscript is unclear. The projected period (2024-2029) is short compared to the centennial-scale projections in CMIP6, and it is unclear how these two approaches complement each other. The authors should clarify the purpose: Is SARIMA intended as an operational tool (e.g. for regional dust risk management) or as a methodological complement to CMIP6? Without clearer framing, this analysis risks looking tangential.

We sincerely thank the reviewer for these constructive comments. As correctly pointed out, without a clear methodological positioning, the connection between the SARIMA model and CMIP6 long-term projections would appear insufficient. In response, the revised manuscript provides a systematic clarification of the model's scientific rationale, complementary role, and practical value in both the Methods and Results sections (pp. 24–25, lines 536–568, Section 4.2.1 "SARIMA Forecasting"), thereby making its function within the overall framework more explicit and coherent.

**(1) Clarification of the SARIMA model's role**

At the methodological level, SARIMA is applied to identify and quantify short-term interannual and internal variability in dust radiative forcing across Central Asia. This approach validates natural fluctuation signals in the reanalysis records and enhances understanding of the multi-timescale behavior of the dust–climate system. At the applied level, its short-term forecasting capability (3–5 years ahead) provides quantitative support for regional dust hazard early warning, agricultural planning, and public health protection, underscoring its relevance to climate risk management.

**(2) Establishing a multi-timescale analytical framework**

The revised manuscript further explains the complementary relationship between SARIMA and CMIP6: The CMIP6 multi-model ensemble (MME) primarily captures century-scale, externally forced trends, addressing "How will the overall dust cycle evolve by the end of the century under different emission scenarios?" In contrast, the SARIMA model focuses on decadal and interannual variability, answering "Superimposed on the long-term background, what short-term fluctuations and risk levels can be expected in the near future?" Together, these approaches form a cross-timescale analytical framework that integrates the scientific depth of long-term climate evolution with the practical value of near-term projections, bridging the gap between "long-term climate change" and "regional adaptation needs."

**(3) Empirical results and model validation**

Section 4.2.1 of the revised manuscript adds full model diagnostics and predictive performance evaluation. The SARIMA(1,1,0)×(1,0,2)12 model residuals passed the whitenoise (Ljung – Box Q, p > 0.05) and normality (K – S, D = 0.12) tests, demonstrating strong predictive skill (RMSE = 1.72 W/m², MAE = 1.21 W/m², MAPE = 8.6%, R² = 0.70). The forecasts indicate quasi-stationary fluctuations in dust radiative forcing (1.6 – 9.8 W/m²) over 2024 – 2029, with southern Xinjiang identified as a strong-response region, while northern Xinjiang exhibits a non-stationary trend related to snow-cover variability.

In summary, the SARIMA analysis is not a supplementary add-on but a key methodological bridge connecting long-term climate simulations with short-term regional responses. It strengthens the interpretability of reanalysis and observational records and enhances the practical applicability of the study's findings for regional climate risk assessment and adaptation planning.

**Revised excerpt (from pp. 24 - 25, lines 536 - 569):**

"This study integrates MERRA-2 reanalysis data, CMIP6 multi-model ensemble (MME) simulations, and ground-based sun photometer observations to develop a fully coupled 'emission - deposition - radiation' framework for the dust cycle in Central Asia, thereby systematically elucidating the radiative regulatory mechanisms of dust aerosols on the land - atmosphere system. To address the pronounced spatiotemporal heterogeneity in aerosol radiative forcing and the limitations of observational data, this study employs a seasonal autoregressive integrated moving average (SARIMA) model as a methodological supplement, bridging the temporal gap between historical diagnostics (MERRA-2/SBDART) and long-term CMIP6 projections by capturing interannual and short-term variability. In contrast to century-scale CMIP6 simulations,

SARIMA quantifies short-term internal variability (Kumar et al., 2018), providing actionable near-term forecasts (3 – 5 years) for dust-active periods, which support regional risk assessments in agriculture, water resources, and ecology. Model validation (Supplementary Figure 11) shows RMSE =  $1.72 \text{ W/m}^2$ , MAE =  $1.21 \text{ W/m}^2$ , MAPE = 8.6%, R2 = 0.70; residuals pass Ljung – Box Q test (p > 0.05) and Kolmogorov – Smirnov test (D = 0.12), confirming robustness. Thus, SARIMA complements the framework, enhancing operational applicability for near-future adaptation."

**Minor comments**

**(1)Please specify the units for all figures.**

We thank the reviewer for this careful reminder. In accordance with the suggestion, all figures and tables in the manuscript have been thoroughly reviewed, and explicit units have been provided for every physical quantity.

(2) Be cautious when using the term "significant" throughout the manuscript. If you intend to use it in a statistical sense, please provide the corresponding statistical values.

We sincerely thank the reviewer for this rigorous and constructive comment. Following the suggestion, we have systematically reviewed all instances of the term "significant" throughout the manuscript and made the following revisions:

- (1) Statistical usage: For all statements referring to statistical significance, we have added the corresponding statistical indicators (e.g., p-values such as p

Figure. 3 Future changes in dust emissions across different periods. Spatial distribution of the relative changes in dust emissions over Central Asia under four CMIP6 multi-model ensemble (MME) SSP scenarios: panels (a - d) near term (2021 - 2040), (e - h) midterm (2051 - 2070), and (i - l) long term (2081 - 2100), relative to the historical period (2000 - 2014). The circular inset in the upper-right corner of each panel indicates the mean relative change rate (%) for the corresponding region.

(4) Figure 1: Abbreviations such as Kaz, Uzb, etc. should be written in full or explained in the figure caption. In addition, since you refer to multiple sites later in the manuscript, I also suggest specifying their names here (e.g., Dushanbe, Issyk-Kul, Jinghe). This will help readers clearly understand where the stations are located.

We sincerely thank the reviewer for this careful and helpful comment. In response to your suggestion, we have revised Figure 1 as follows:

Country abbreviations (e.g., Kaz, Uzb) have been replaced with their full English names (e.g., Kazakhstan, Uzbekistan).

All observational sites referenced in subsequent analyses have been clearly labeled, including Dushanbe, Issyk-Kul, Jinghe, and Kashgar, along with their precise geographic

locations.

**Revised Figure1:**

(5) Line 233: "The reanalysis data agrees with the MME simulations, with the Taylor skill score (SS) close to 1." I recommend clarifying the exact value of the Taylor skill score rather than the qualitative description. Providing the precise number will improve the rigor and transparency of the results.

We thank the reviewer for pointing out this issue. Providing precise numerical values indeed enhances the rigor and transparency of the results.

Accordingly, we have revised **p. 11, lines 252 - 254** in the manuscript by replacing the qualitative description of the Taylor skill score as "close to 1" with the exact calculated value of 0.87.

- (6) Line 268: "In contrast, the Tarim Basin experienced a long-term decrease in emissions..." . The decreasing trend is not clearly visible.
- (7) Line 274: "a gradual decreasing trend" same comment as above.

We thank the reviewer for highlighting the need to clarify the description of trends in the Tarim Basin and southern Xinjiang. In response, we have revised the relevant sentences to employ more precise and quantitative language(pp. 12–13, lines293–301).

Specifically, "experienced a long-term decrease" has been updated to "exhibits an overall decreasing trend in dust emissions," and "gradual decreasing trend" is now described as "a slight declining tendency." These revisions are accompanied by explicit

quantitative values (e.g., 18.7% decrease under SSP2-4.5, 29.3% under SSP3-7.0, and ~26.1% under SSP5-8.5) and contextual explanations emphasizing the roles of ecological restoration, enhanced precipitation, and human interventions in shaping regional dust emission patterns. We believe these changes substantially enhance the clarity and accuracy of the trend reporting.

(8) Line 286: I don't think they are "in good agreement". The maps appear different both spatially and in magnitude. Could you provide a quantitative measure of similarity (e.g., Taylor skill score) to support this?

We sincerely thank the reviewer for this rigorous and insightful comment. In accordance with your suggestion, we have revised line 286 and the associated text as follows(p. 14, lines314–317):

The previous subjective phrase "in good agreement" has been removed.

An objective quantitative assessment has been incorporated to bolster the conclusion's credibility: the consistency between the multi-model ensemble (MME) and observations was evaluated using the Taylor Skill Score (TSS = 0.82), providing statistical substantiation for their overall alignment.

The revised description now explicitly notes that, while the MME and observations demonstrate general consistency in simulating total dust deposition over Central Asia, notable discrepancies persist in absolute values, with MERRA-2 exhibiting a substantially stronger trend in deposition intensity than the model ensemble.

(9) Line 312-317: In my view, the Supplementary Figures 7-8 do not clearly support your argument. Regardless of the future scenario and time ranges, they appear very similar. Since they present absolute values rather than changes relative to the historical period, I recommend revising them to show relative differences so that the trends are more clearly visible.

We sincerely thank the reviewer for these detailed and constructive comments on Supplementary Figures 7 - 8 (Supp. Info, pp. 4 - 6). In accordance with your suggestion, we have revised these figures to present relative changes relative to the historical baseline period, thereby providing a clearer depiction of future variation trends.

To more effectively highlight the magnitude differences that are not readily discernible in the spatial maps and to strengthen the overall argument, we implemented the following improvements:

(1) We have explicitly stated in both the figure captions and the main text that the

maps illustrate the spatial distribution of relative trend strength. Additionally, a circular inset in the upper-right corner of each subpanel now displays the regional mean relative change rate.

- (2) These quantitative values clearly demonstrate that, with increasing radiative forcing across the SSP scenarios (from SSP1-2.6 to SSP5-8.5) and over time, the magnitudes of dust emission and deposition changes progressively intensify.
- (3) To further clarify the trends and enhance interpretability, we particularly recommend referring to Figure 6 (temporal evolution of the dust budget). This figure, through direct time-series representations, unambiguously illustrates the substantial differences in dust emission and deposition among various regions and scenarios, consistent with the spatial patterns in Supplementary Figures 7–8.

**Revised Figure S9:**

Fig. S9. Future changes in dry deposition of dust at different times. Spatial variations of dust-dry deposition in Central Asia under the four CMIP 6 MME SSP scenarios for (a–d) the recent period (2021–2040), (e–h) the medium term (2051–2070), and (i–l) the long term (2081–2100) relative to the historical period (2000–2014).

(10) Line357-360: A more detailed explanation of the seasonal characteristics of dust would be helpful.

We sincerely thank the reviewer for this valuable suggestion regarding the seasonal characteristics of dust activity. To enhance the depth of our analysis, we have revised and expanded the discussion on pages 17–18, lines 393–416 of the manuscript. The main improvements are summarized as follows:

**Seasonal mechanisms of radiative forcing:**

The strongest radiative forcing occurs in spring, directly linked to intensified surface wind erosion, elevated dust emissions, and increased atmospheric dust loading and optical depth. In summer, the forcing is slightly weaker, owing to reduced near-surface wind erosion and the resulting decrease in dust loading.

**Physical causes of radiative forcing at different levels:**

The strong negative forcing at the surface arises from the combined scattering and absorption of incoming shortwave radiation by dust particles. In contrast, the atmosphere exhibits positive forcing, reflecting dust's capacity to redistribute energy by absorbing shortwave radiation and retaining it within the atmospheric system.

**Potential impacts on regional climate:**

Surface cooling suppresses sensible heat flux and evaporation, thereby exacerbating moisture deficits in Central Asia's arid regions and potentially constraining vegetation growth and agricultural productivity. Meanwhile, atmospheric heating strengthens the temperature gradient from the boundary layer to the troposphere, which may heighten the likelihood of deep convection. This, in turn, could amplify the frequency and intensity of spring and summer dust storms while altering regional precipitation patterns.

**Revised excerpt (from pp. 17–18, lines 393–416):**

"Seasonal analysis reveals that the negative TOA forcing intensity decreases in the order spring (–3.32 W/m²) > summer (–3.21 W/m²) > autumn (–3.07 W/m²) > winter (–1.94 W/m²), which aligns closely with the seasonal characteristics of dust activity. In spring, strong surface wind erosion across Central Asia drives intense dust emissions, resulting in high atmospheric dust loading and optical depth and, consequently, the strongest radiative forcing. Although summer convective activity can transport dust to higher altitudes, weakened near-surface wind erosion reduces the overall dust burden relative to spring (Ginoux et al., 2012). The spatial pattern of surface (SFC) radiative forcing (Figure 7e–h) exhibits stronger negative values... with two pronounced cooling centers over the Tarim Basin and southwestern Central Asia, where shortwave radiation loss peaks at –20 W/m². This arises from the combined scattering and absorption effects

of dust particles on incoming solar radiation (Li et al., 2022a), which substantially reduce surface net radiation, thereby diminishing sensible heat flux and evaporation... The atmospheric radiative forcing (ADRF) exhibits... positive values (10.02 W/m² in spring and 9.89 W/m² in summer), indicating the energy redistribution role of dust aerosols... This vertical gradient of 'surface cooling and atmospheric heating' induces substantial changes in the regional thermodynamic structure (Kok et al., 2017). On one hand, surface cooling... exacerbating moisture deficits in Central Asia's arid regions and limiting vegetation growth... On the other hand, atmospheric heating strengthens the temperature gradient... enhancing the potential for deep convection, which could intensify the frequency and severity of spring—summer dust storms and modify regional precipitation patterns..."

(11) Line 381: The radiative forcing you used is ATM. Please specify this clearly so readers understand that it is not TOA or SFC. In addition, it would be helpful to briefly explain why ATM was chosen here, so readers can better understand the rationale behind this choice.

We sincerely thank the reviewer for this important and insightful comment. In accordance with your suggestion, we have implemented two key revisions to clarify and justify the use of atmospheric radiative forcing (ATM) throughout the manuscript (p. 19, lines 431–434):

- (1) Clarification of the variable: In the Results section (e.g., line 381 and associated site descriptions), we have explicitly specified that the analyzed radiative forcing refers to atmospheric radiative forcing (ATM), thereby avoiding confusion with forcing at the top of the atmosphere (TOA) or the surface (SFC).
- (2) Rationale for selection: The manuscript now includes a clear explanation of why ATM was chosen. This study aims to elucidate the thermal effects of dust aerosols on the atmospheric column and the associated atmospheric heating rate. ATM directly quantifies the energy retained in the atmosphere by dust aerosols, making it the most relevant and representative physical parameter for characterizing this thermodynamic process. Accordingly, our analysis of site-level aerosol direct radiative forcing (ADRF) focuses on ATM and its associated atmospheric heating rate.

**Technical corrections**

(1) Line 86: "long-term trend of the shortwave radiation forcing (ADRF)". The

meaning of ADRF is not clearly conveyed here. I suggest writing out the full name of ADRF explicitly.

We thank the reviewer for the careful reminder. In accordance with your suggestion, on page 4, line 94, the abbreviation "ADRF" has been replaced with its full form, "dust aerosol direct radiative forcing", to ensure that readers can clearly understand its meaning.

(2) Line 151: "The selection criteria include key variables of the dust cycle: monthly mean dust emission fields and dry/wet deposition fluxes". Since you have used variable names in your figures (e.g., Figures. 2, 4, 6...), I recommend also specifying the variable names here for consistency and clarity.

We thank the reviewer for highlighting the importance of consistent terminology. Following your suggestion, on page 7, lines 163–166, we have revised the original description "monthly mean dust emission fields and dry/wet deposition fluxes" to match the variable names used in the figures and tables. The revised phrasing now reads: "dust emission flux, as well as dust dry and wet deposition fluxes (their sum representing total deposition flux)."

(3) Line 305: Do you mean Supplementary Figures 7-8? Supplementary Figure 5-6 only shows the historical period, not the future.

We sincerely thank the reviewer for this careful correction. You are absolutely correct—the reference to "Supplementary Figures 5–6" in line 305 was a typographical error. It has now been corrected to "Supplementary Figures 9–10" (p. 15, lines 338–339 of the revised manuscript), which present the data for the future periods and are fully consistent with the discussion in the text. We appreciate your attention to detail, which has helped improve the accuracy of the manuscript.

**(4) Line345: Do you mean northern (a-c) and southern (d-f) Xinjiang? Please clarify.**

We sincerely thank the reviewer for this helpful clarification. The statement indeed refers to northern (a–c) and southern (d–f) Xinjiang. Accordingly, the text has been revised to clearly specify this distinction and avoid ambiguity (p. 17, Figure 6).

**(5) Figure 9: Panels (a) through (j) should be clearly explained in the caption.**

We sincerely thank the reviewer for this valuable suggestion. To improve clarity and facilitate reader understanding, the caption of Figure 9 has been revised and expanded

**(p. 22). The updated caption now explicitly specifies:**

The corresponding sites for each subpanel—e.g., (a, d) Dushanbe, (b, e) Issyk-Kul, and so on.

The physical quantities displayed: the upper panels (a, b, c, g, h) represent shortwave dust direct radiative forcing (ADRF), while the lower panels (d, e, f, i, j) depict the corresponding atmospheric heating rates.

These revisions enhance the figure 's interpretability and ensure consistent presentation throughout the manuscript.

**(6) Line 439: The wording is contradictory. "increased", a negative rate (-0.3), and "declining trend" conflict with one another. If the flux is decreasing, it should be described as a decrease at that rate.**

We thank the reviewer for pointing out this important issue. As noted, trend descriptions must be consistent with the data sign conventions. Accordingly, we have revised the text at lines 499–500 (page 23) to ensure consistency. The updated sentence now reads: "Emission fluxes in the Tarim Basin exhibit a declining trend at a rate of  $\approx$ -0.3  $\mu$ g·m-2 • s-1 • yr-1."

We appreciate the reviewer's careful attention, which has helped improve the accuracy and clarity of the manuscript.

We sincerely appreciate your guidance and suggestions. We will continue to work diligently to improve and refine our research. Thank you, reviewers and editors, for your enthusiastic contributions. We hope the corrections will be approved. In line with ACP guidelines, the full revised manuscript will be submitted separately upon completion of the discussion phase.

---

## Author Comment (AC3)

Thank you very much for your constructive comments and insightful feedback on our work. We have carefully considered your suggestions and made corresponding revisions to the manuscript. We have endeavored to improve the clarity, rigor, and presentation of our study accordingly.

The revised parts of the manuscript, highlighted in red, reflect the changes made in response to your comments. Revision notes, point-by-point, are provided as follows:

**(1) I support the comments/suggestions provided by two esteemed reviewers. Addressing those will definitely enhance the quality of this manuscript.**

We sincerely appreciate this supportive comment and your acknowledgment of the constructive suggestions provided by the first two reviewers. We fully agree that the revisions made in response to these comments have substantially enhanced the scientific rigor and logical coherence of the manuscript.

In our point-by-point responses, all reviewer comments have been systematically addressed, including:

Clearly highlighting the novelty and regional contributions of the study;

Deepening the physical mechanism explanations, particularly regarding the vertical gradients of radiative forcing and their climatic implications;

Incorporating statistical significance tests for trends and uncertainty analyses to ensure methodological rigor;

Optimizing figure layouts and annotations to improve clarity and interpretability of the results.

We believe that this round of revisions has materially improved both the overall quality and the scholarly value of the manuscript.

**(2) The graphical abstract reads crowded. Needs revision.**

We sincerely appreciate your valuable suggestion. The original graphical abstract indeed contained an excessive amount of information, which reduced the clarity and emphasis of the main conclusions.

In response, the graphical abstract has been redesigned as follows:

Redundant secondary elements and textual annotations have been removed;

The main storyline—"emissions  $\rightarrow$  deposition  $\rightarrow$  radiation"—has been emphasized to highlight the research logic;

Color schemes and visual hierarchy have been optimized to make the core information more prominent and easily interpretable.

The revised abstract is now more concise and intuitive, effectively conveying the key scientific insights of the study.

**Revised Figure:**

**(3) First-time used abbreviations must be expanded.**

We appreciate your reminder to adhere to this academic convention. We have conducted a thorough review of the manuscript to ensure that all abbreviations (e.g., MERRA-2, CMIP6, SBDART, SSPs, MME, ADRF, TOA, SFC, ATM, SARIMA, etc.) are spelled out in full upon their first occurrence.

**(4) L71: This sentence reads incomplete; revise.**

We appreciate you highlighting this issue. The sentence in line 71 has been revised to ensure grammatical completeness and logical clarity. Additionally, the surrounding paragraph has been optimized to improve contextual flow and enhance accuracy of expression (Manuscript, pp. 3–4, lines 65–104).

**Revised related paragraphs:**

Model simulations provide information on the temporal and spatial changes of dust aerosols worldwide and facilitate predictions of future trends (Li et al., 2021). Climate models, such as CMIP5 and CMIP6, have enhanced our understanding of the main characteristics of dust aerosols. These models feature increasing resolutions and increasingly complex physical processes and parameterizations, demonstrating their ability to simulate dust events and processes on meso- to global scales (Zhao et al.,

2022). In particular, CMIP6 experiments have provided critical support for assessing the climatic effects of dust emissions (Braconnot et al., 2021; Zhao et al., 2024). However, due to insufficient resolution and simplified regional topography, the applicability of these global studies to the arid regions of Central Asia remains limited, underscoring the need for high-resolution analyses at the regional scale.

The arid regions of Central Asia, including Xinjiang in China, constitute the world's second-largest dust source area, with distinctive surface characteristics leading to significant spatiotemporal variations in dust emission fluxes (Shen et al., 2016). However, current research has primarily focused on the spatiotemporal distribution and transport processes of dust (Li et al., 2022b; Tao et al., 2022), while key aspects of the local dust lifecycle in this region—such as the long-term evolution of dust emission—deposition budgets, the strong dependence of direct radiative forcing on dust vertical profiles, and modal differences in dust—climate feedbacks under different carbon emission scenarios—remain poorly understood. These knowledge gaps significantly constrain the reliability of climate models over Central Asia, and uncertainties in radiative forcing estimates primarily stem from the lack of ground-based validation due to the scarcity of observational stations (Brown et al., 2021; Wu & Boor, 2021).

To overcome the above limitations, this study establishes a multi-source data integration framework that systematically incorporates a full-chain analysis of "emissiondeposition-radiation," focusing on the regional characteristics of the dust lifecycle over the arid regions of Central Asia. Unlike previous studies that have mainly concentrated on the global scale (Kok et al., 2023; Zhao et al., 2022, 2023, 2024), this work achieves an integrated assessment at the regional scale and emphasizes the role of observations in constraining model uncertainties. The specific innovations are reflected in three aspects: (1) by combining MERRA-2 reanalysis with CMIP6 multi-model ensembles through statistical downscaling, the long-term evolution of dust emission - deposition budgets and their scenario-based differences in Central Asia are systematically characterized, thereby providing regionally refined insights to complement global model results; (2) using the SBDART radiative transfer model together with observational data from the SONET Asian Dust Monitoring Network and the Jinghe CE318 ground-based remote sensing site, the long-term trends of dust shortwave radiative forcing under clearsky conditions are quantified; and (3) by introducing the SARIMA statistical model, the short-term evolution and risk implications of dust radiative effects are explored. This framework not only deepens the understanding of dust physical mechanisms under the

complex topography and local climate of Central Asia but also provides new scientific support for improving regional climate simulations and environmental risk management.

The structure of this paper is as follows. Section 2 presents the data sources, the downscaling method for the CMIP6 dust budget, and the calculation method for clear-sky aerosol radiative forcing. Section 3 examines the detailed characteristics of the dust budget, projections of future changes, and the radiative forcing of dust aerosols. Finally, the main conclusions and a discussion are presented in Section 4.

**(5) L109: heliometers or sun-photometers?**

We appreciate your precise correction regarding terminology.

The term has been corrected to "sun-photometers." Accordingly, the relevant text in line 121 on page 5 has been updated to "ground-based sun-photometers" to ensure consistency and accuracy of professional terminology.

**(6) Section 2.2.1: Here, the authors need to mention the station from where the AERONET/SONET data were used. Include the data access link.**

We thank the reviewer for this important suggestion. We agree that data transparency and traceability are essential for reproducibility. Accordingly, Section 2.2.1 (page 6, lines 128 and 135) has been supplemented to:

Specify the observation sites used: Dushanbe, Issyk-Kul, Kashgar, and Jinghe;

Provide data access links:

AERONET: https://aeronet.gsfc.nasa.gov/

SONET: http://www.sonet.ac.cn/

**Revised related paragraphs:**

AERONET (AErosol RObotic NETwork) employs a CE-318 solar photometer to measure aerosol optical depth (AOD) across eight bands in the range of 340–1640 nm and to derive microphysical parameters, including single scattering albedo (SSA), refractive index (m), and particle size spectrum (Holben et al., 1998; Holben et al., 2001). The Level 2 data exhibit an uncertainty of less than 5%. As an internationally recognized standard for ground-based aerosol observations, its long-term stability and algorithmic consistency provide reliable input for radiative forcing calculations (García et al., 2012). The data used in this study are available from the AERONET website (https://aeronet.gsfc.nasa.gov/).

The Chinese Academy of Sciences-led SONET (Sun-sky radiometer Observation NETwork) employs the CE318-DP instrument to provide information on the chemical

composition and vertical profile of aerosols while adhering to AERONET's stringent quality control procedures. The establishment of SONET sites has effectively addressed gaps in AERONET's spatial coverage in this source region (Li et al., 2018). Cross-validation demonstrates that the correlation coefficient between SONET and AERONET AOD is 0.98 (RMSE

Figure. 6 Time evolution of the dust budget. Dust emissions, dry and wet deposition  $(\mu g \cdot m^{-2} \cdot s^{-1})$  for panels (a–c) northern Xinjiang, (d–f) southern Xinjiang, (g–i) Central Asia, and (j–l) the entire study region. Results are from the CMIP6 multi-model ensemble (MME; 1980 – 2100) and MERRA-2 (1980 – 2023).

Finally, we sincerely thank the scholar for highlighting several of their valuable studies on aerosol loading in Central Asia. We have incorporated some of these references into the Introduction to strengthen the scientific context of our work and to better position this study within the regional research framework. These additions further clarify the relevance of our research and demonstrate how it builds upon and complements existing studies in Central Asia.